# DINGO: increasing the power of locus discovery in maternal and fetal genome-wide association studies of perinatal traits

Liang-Dar Hwang [1]✉, Gabriel Cuellar-Partida[2], Loic Yengo [1], Jian Zeng[1], Jarkko Toivonen [3], Mikko Arvas [3], Robin N. Beaumont [4], Rachel M. Freathy [4,5], Gunn-Helen Moen[1,6,7,8], Nicole M. Warrington [1,5,7,8] & David M. Evans [1,5,8]✉

Perinatal traits are influenced by fetal and maternal genomes. We investigate the performance of three strategies to detect loci in maternal and fetal genome-wide association studies (GWASs) of the same quantitative trait: (i) the traditional strategy of analysing maternal and fetal GWASs separately; (ii) a two-degree-of-freedom test which combines information from maternal and fetal GWASs; and (iii) a one-degree-of-freedom test where signals from maternal and fetal GWASs are meta-analysed together conditional on estimated sample overlap. We demonstrate that the optimal strategy depends on the extent of sample overlap, correlation between phenotypes, whether loci exhibit fetal and/or maternal effects, and whether these effects are directionally concordant. We apply our methods to summary statistics from a recent GWAS meta-analysis of birth weight. Both the two-degree-of-freedom and meta-analytic approaches increase the number of genetic loci for birth weight relative to separately analysing the scans. Our best strategy identifies an additional 62 loci compared to the most recently published meta-analysis of birth weight. We conclude that whilst the two-degree-of-freedom test may be useful for the analysis of certain perinatal phenotypes, for most phenotypes, a simple meta-analytic strategy is likely to perform best, particularly in situations where maternal and fetal GWASs only partially overlap.

Perinatal traits like birth weight are influenced by genetic variants from both offspring and maternal genomes[1–6]. Historically, genome-wide associations studies (GWASs) of these traits have involved separate analyses of offspring and maternal genomes, i.e., a fetal GWAS where an individual's own phenotype is regressed on their own genotype, and a separate maternal GWAS where offspring phenotype is regressed on maternal genotype. Genetic variants showing genome-wide significant associations in either scan are then followed up using conditional association analyses (or transmitted and non-transmitted haplotype analyses) to investigate whether their effects are due to the fetal genotype, the maternal genotype, or some combination of both (NB. In this manuscript, we refer to the effect of the maternal genotype on the

[1]Institute for Molecular Bioscience, The University of Queensland, St Lucia, Australia. [2]Gilead Sciences, Inc, Foster City, CA, USA. [3]Finnish Red Cross Blood Service, Vantaa, Finland. [4]Department of Clinical and Biomedical Sciences, Faculty of Health and Life Sciences, University of Exeter, Exeter, UK. [5]MRC Integrative Epidemiology Unit, University of Bristol, Bristol, UK. [6]Institute of Clinical Medicine, Faculty of Medicine, University of Oslo, Oslo, Norway. [7]Department of Public Health and Nursing, K.G. Jebsen Center for Genetic Epidemiology, NTNU, Norwegian University of Science and Technology, Trondheim, Norway. [8]The Frazer Institute, The University of Queensland, Woolloongabba, QLD, Australia. ✉e-mail: d.hwang@uq.edu.au; d.evans1@uq.edu.au

offspring phenotype as an indirect maternal genetic effect since the effect is mediated indirectly through the intrauterine environment. In contrast, we refer to the effect of an individual's own genotype on their own phenotype (here birthweight), as a direct fetal genetic effect[6,7]. Using this strategy, large-scale GWAS meta-analyses have identified over 280 variants robustly associated with a range of perinatal traits through both the fetal and/or maternal genomes[1–6,8–10].

Whilst this strategy of conducting separate fetal and maternal GWASs has been successful in terms of locus identification, it is not optimal statistically, because it does not utilize information shared across the individual GWASs. Since maternal and offspring genotypes are correlated, a GWAS of an individual's own phenotype also provides information on indirect maternal genetic effects (i.e. in addition to information on direct fetal genetic effects). Likewise, GWASs of offspring phenotype also provide information on direct fetal genetic effects (i.e. in addition to indirect maternal genetic effects). This issue is most pronounced when the fetal and maternal GWASs are only partially overlapping (and hence contain independent information)- which has certainly been the case in the past for the GWAS meta-analysis of many phenotypes within the Early Growth Genetics (EGG) consortium including birth weight[1–6]. However, it may not be clear how best to combine information from maternal and fetal GWASs, particularly when the degree of sample overlap/relatedness is unknown. In addition, when conducting two or more GWASs (e.g. a GWAS of one's own trait and then a GWAS of the same trait in one's offspring), investigators should ideally increase the statistical penalty due to multiple testing- although this is not often done in practice and may not be easy to do optimally given

unknown sample overlap and correlation between the traits. Data simulations and asymptotic power calculations we have performed previously have hinted at the gains in power that might be achieved by simultaneously modeling indirect and direct effects[7,11].

In this work we examine the performance of a computationally simple two-degree-of-freedom test that uses GWAS summary results statistics from fetal and maternal GWASs. Our method uses LD score regression[12] to estimate an effective sample overlap across fetal and maternal GWASs, and then utilizes this information to estimate the sampling variance and covariance of the conditional direct fetal and indirect maternal genetic effect estimates. In addition, by essentially performing a single GWAS, the method avoids the thorny issue of how best to adjust for multiple testing across two correlated GWASs that contain an unknown proportion of overlapping/related individuals. We also compare the two-degree-of-freedom test to a simple one-degree-of-freedom test where signals from maternal and fetal GWASs are meta-analysed together conditional on the estimated sample overlap, as well as to the traditional strategy of separately analysing maternal and fetal GWASs. We investigate the power of the different approaches through a combination of analytical formulae and data simulation and provide users with a web tool to conduct their own asymptotic power calculations. We apply the different methods to summary statistics from the most recent GWAS of birth weight involving the deCODE Study, EGG Consortium and UK Biobank[5] and illustrate how combining the data dramatically increases the number of significantly associated loci for these traits relative to analysing the data separately or via the commonly used MTAG (Multi-trait analysis of GWASs) program[13]. We show how our

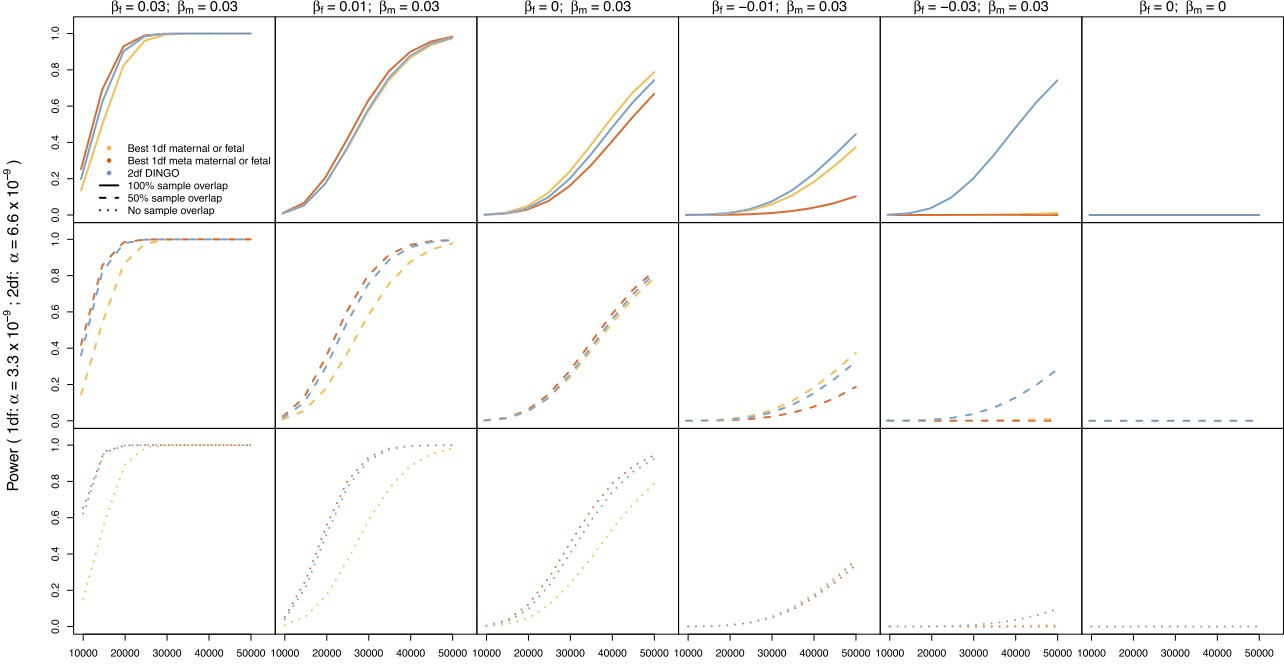

**Fig. 1 | Power to detect association (as evaluated asymptotically) using a traditional strategy of analysing separate maternal and fetal GWASs, a strategy involving one-degree-of-freedom meta-analyses, and a strategy of performing two-degree-of-freedom $T_{2df}$ tests across the genome.** $\beta_m$ and $\beta_f$ refer to maternal and fetal genetic effects on a standardized trait. In the top row, results are shown for replicates where there is complete sample overlap, and in the middle row, 50% sample overlap. Data were simulated assuming a high residual correlation between maternal and offspring phenotypes ($\rho = 0.5$). In the bottom row, results are shown for replicates where there is no sample overlap. For the traditional strategy of

running separate maternal and fetal GWASs and the one-degree-of-freedom meta-analysis strategy, we set the alpha value to $\alpha = 3.3 \times 10^{-9}$, i.e. half the $\alpha = 6.6 \times 10^{-9}$ type I error rate of the two-degree-of-freedom $T_{2df}$ test in order to take into account that we are performing twice the number of statistical tests in the former situations. In the case of the traditional strategy of running separate maternal and fetal GWASs and the one-degree-of-freedom meta-analysis strategy, we evaluated power with respect to whether *either* test met the criterion for genome-wide significance for each replicate.

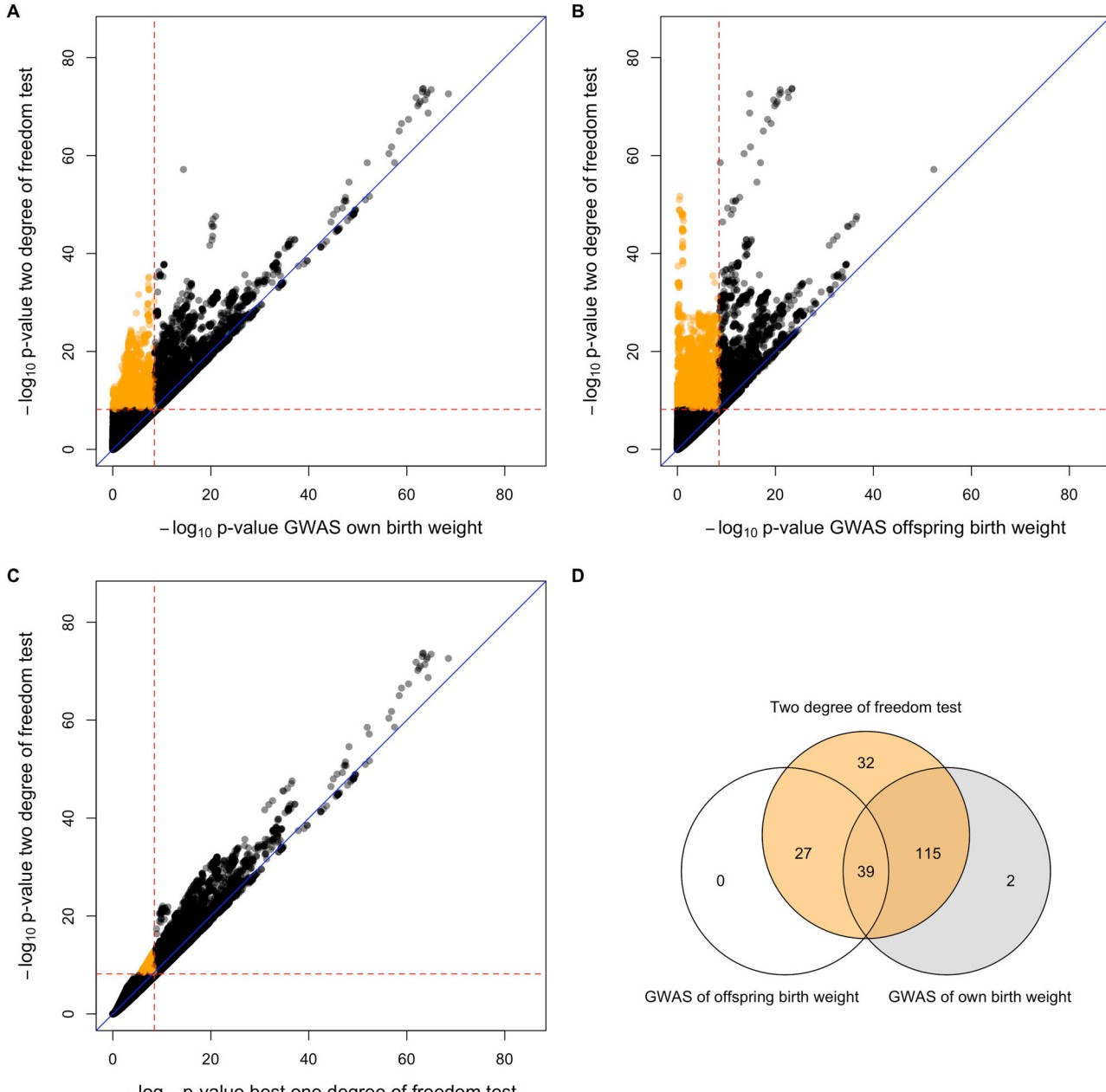

**Fig. 2 | *P*-value scatter plots and Venn diagram comparing results from separate GWASs of own and offspring birth weight versus the combined two-degree-of-freedom test across 31,033,794 SNPs from the deCODE study of birth weight.** The -log$_{10}$*p*-value from the two-degree-of-freedom test (*y*-axis) was compared against the -log$_{10}$*p*-value of SNPs from (**A**) the separate GWAS of own birth weight, (**B**) the separate GWAS of offspring birth weight, and (**C**) the stronger *p*-value from among **A** and **B** (x-axis). Red dashed lines denote genome-wide significant thresholds of $\alpha = 3.3 \times 10^{-9}$ for the separate GWAS of own birth weight and offspring birth weight which involve one-degree-of-freedom tests, and $\alpha = 6.6 \times 10^{-9}$ for the two-degree-of-freedom test. The blue diagonal lines indicate x = y. Orange circles are SNPs that are only genome-wide significant in the two-degree-of-freedom test. **D** illustrates the degree of overlap across genome-wide significant loci identified from the two strategies.

methods can be extended to investigate the genetic etiology of other phenotypes like educational attainment and IQ that are putatively influenced by indirect maternal and indirect paternal genetic effects as well as the direct effect from individuals' own genomes[14–16]. Finally, we implement our methods in the DINGO (**D**irect and **IN**direct effects analysis of **G**enetic l**O**ci) software package, part of the online Complex Trait Genetics Virtual Laboratory (CTG-VL)(https://vl.genoma.io/)[17], which allows users to perform these tests easily and computationally efficiently across the genome using summary results GWAS data.

## Results

### Power analyses

Figure 1 and Supplementary Figs. 1–5 compare the power to detect association across the three analytic strategies examined in this manuscript. When there was minimal sample overlap between maternal and fetal GWASs, then the usual strategy of analysing maternal and fetal GWASs separately was typically less powerful than strategies that combined information from both GWASs (Fig. 1 bottom panel). In general, in these situations, the meta-analysis strategy was often slightly more powerful than the two-degree-of-freedom test in

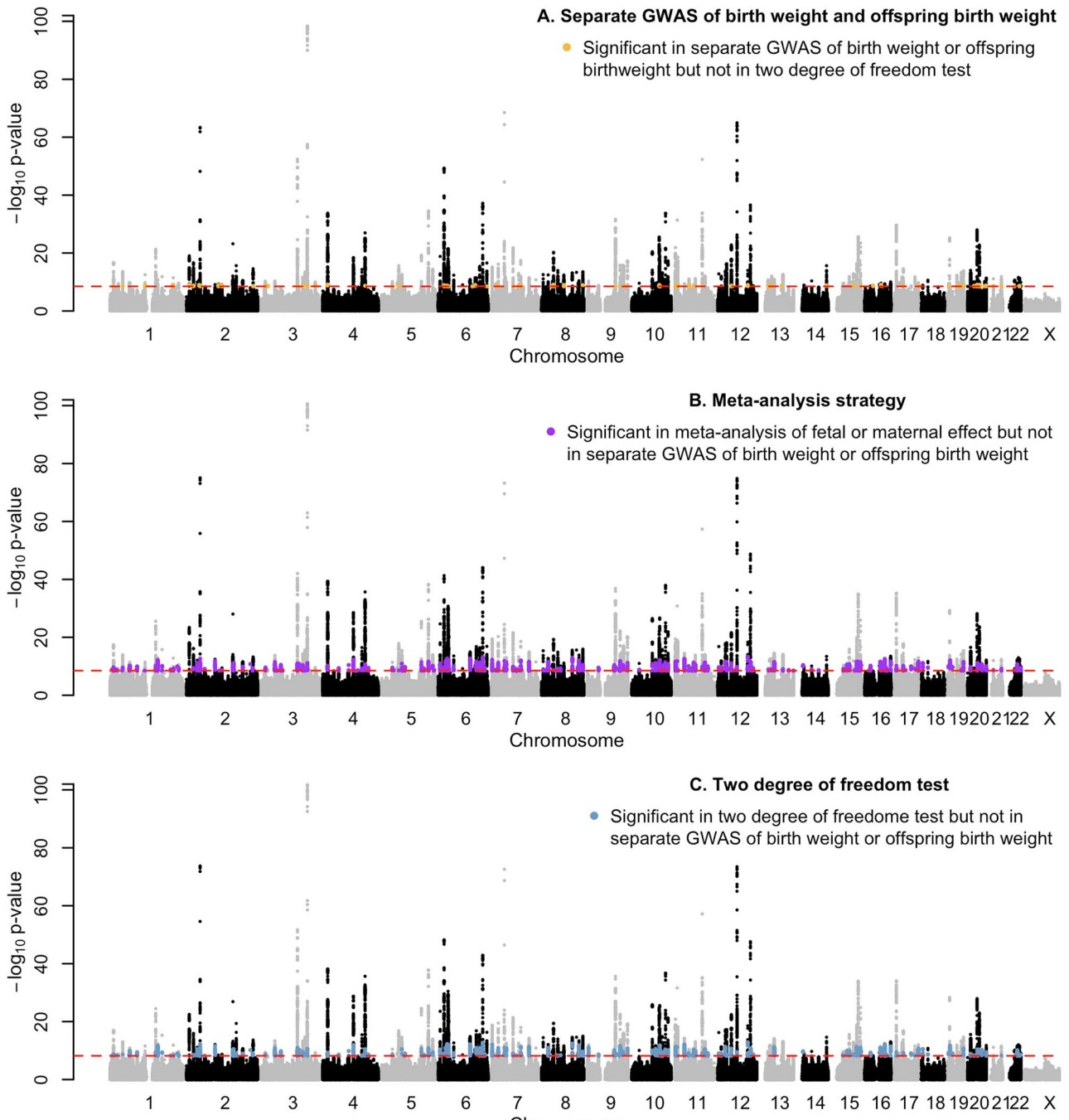

**Fig. 3 | Manhattan plots showing results of using the typical strategy of conducting separate GWASs of birth weight and offspring birth weight. a** The typical strategy of conducting separate GWASs of birth weight and offspring birth weight (the best p-value at each locus shown). **b** A simple meta-analytic strategy (the best p-value at each locus shown). **c** The two-degree-of-freedom test. The -log p-value for the SNP association is plotted on the y-axis. Red dashed lines indicate genome-wide significant thresholds of p-value = $3.3 \times 10^{-8}$ for one-degree-of-freedom tests, and p-value = $6.6 \times 10^{-8}$ for the two-degree-of-freedom tests. Orange points represent SNPs that are genome-wide significant in the separate GWASs of birth weight/offspring birth weight, but not in the two-degree-of-freedom test. Purple points represent SNPs that are genome-wide significant in the metaanalysis of fetal or maternal effects, but not in the separate GWAS of birth weight/offspring birth weight. Blue points represent SNPs that are genome-wide significant in the two-degree-of-freedom test, but not in the separate GWAS of birth weight/offspring birth weight.

detecting loci that exhibited one type of effect only and when detecting loci with concordant maternal and fetal effects.

In contrast, at the other extreme, when there was complete sample overlap, then the optimal strategy depended upon whether the locus exhibited both types of effects concurrently and the residual correlation between variables. For example, Fig. 1 shows an instance where the usual strategy of analysing the maternal and fetal

GWASs separately was most powerful- in this case where loci exhibited only indirect maternal (or only direct fetal) genetic effects and in which the residual correlation was very high. In contrast, when loci exhibited concordant maternal and fetal effects of similar magnitude then both the two-degree-of-freedom test and the meta-analysis strategy often gave superior power compared to separately analysing the scans.

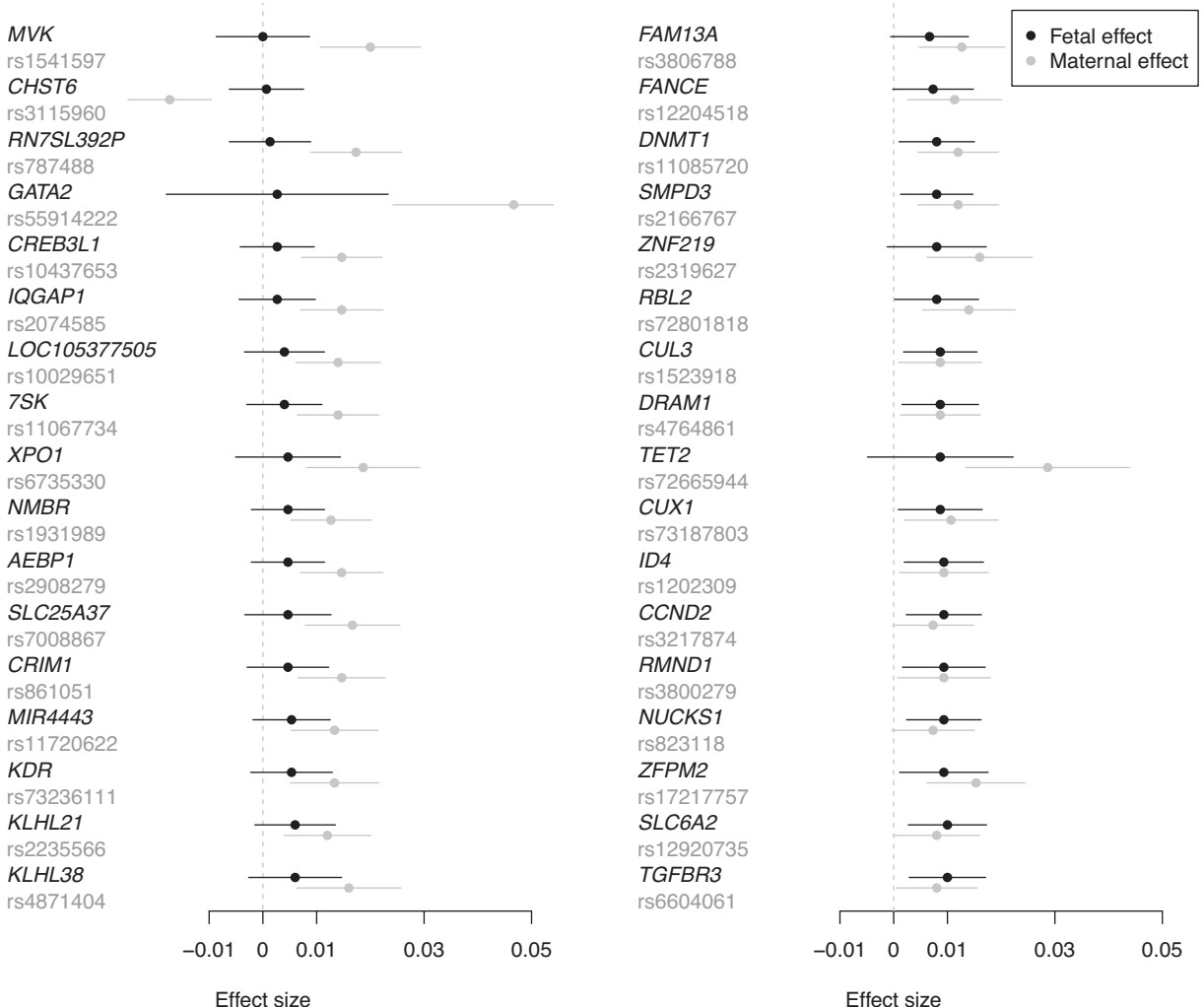

**Fig. 4 | Estimated conditional fetal and maternal effects and 95% confidence intervals for the 34 independent genome-wide significant SNPs identified in the two-degree-of-freedom analysis ($p$-value $< 6.6 \times 10^{-9}$) that did not meet genome-wide significance in the separate GWAS of birth weight ($n = 423,683$)** **and offspring birth weight ($n = 270,002$) ($p$-value $< 3.3 \times 10^{-9}$).** Effects are oriented to the allele that is associated with an increased fetal effect on birth weight. The SNPs are labeled according to the physically closest gene. The upper 95% confidence interval for rs55914222 has been truncated.

Interestingly, in the case of loci that exhibited discordant maternal and fetal effects (of similar magnitude), the two-degree-of-freedom $T_{2df}$ test provided increased power to detect association relative to the other strategies, particularly in overlapping samples. Notably, the simple meta-analysis strategy often did not perform well in these situations (see Supplementary Data 1 for further details).

In the Supplementary Note, we derive asymptotic formulae for the unconditional GWAS tests, the meta-analytic strategy, and the two-degree-of-freedom tests of association. We confirm that our formulae for asymptotic power of the two-degree-of-freedom test derived in the Supplementary Note gave similar results to those obtained using simulated data (Supplementary Data 1) and implement our calculations in a web calculator (https://evansgroup.di.uq.edu.au/DINGO) that investigators can use to investigate power in their own studies.

**Empirical analyses of birth weight using the two-degree-of-freedom test**

The GWAS of birth weight using the two-degree-of-freedom $T_{2df}$ test identified 332 independent SNP associations at 213 loci with a $p$-value $< 6.6 \times 10^{-9}$ (Figs. 2 and 3 and Supplementary Data 2). This included 34 independent SNPs at 32 loci that did not reach genome-

wide significance in either the separate GWASs of birth weight or offspring birth weight (Supplementary Data 2). These results are consistent with the preceding power calculations which suggest that performing a two-degree-of-freedom test can be substantially more powerful than performing separate maternal and fetal GWASs when there is only partial overlap between the GWASs. Conditional analyses suggested that among 34 genome-wide significant SNPs at the 32 loci, 10 acted primarily through the fetal genome, 12 acted primarily through the maternal genome, and 12 acted through both ($p$-value $< 0.05$) (Fig. 4). Interestingly, of the 12 SNPs that showed evidence for both maternal and fetal effects in the conditional analyses, all 12 SNPs exhibited effects in the same direction (Fig. 4). These results were at first surprising, given that the power calculations from the previous section suggested that whilst the two-degree-of-freedom test should increase the power to detect loci that show concordant maternal and fetal effects when there is partial sample overlap, the largest gains in power should involve the detection of SNPs that have discordant maternal and fetal effects of similar magnitude. Indeed only 22 of 332 genome-wide significant SNPs identified using the two-degree-of-freedom test exhibited conditional maternal ($p$-value $< 0.05$) and fetal effects ($p$-value $< 0.05$) in opposite directions- and all these SNPs were also identified in the separate GWAS analyses of own

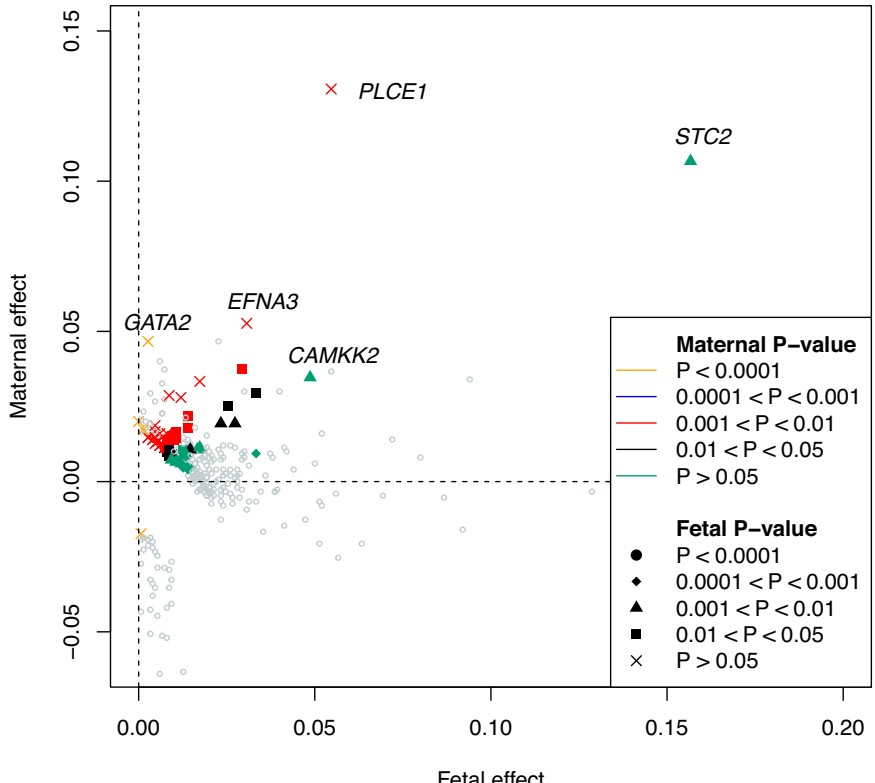

**Fig. 5 | Estimates of conditional fetal and maternal effects for 332 genome-wide significant SNPs identified in the two-degree-of-freedom GWAS.** 76 SNPs* that reached genome-wide significance in the two-degree-of-freedom test ($p$-value $< 6.6 \times 10^{-9}$), but not in either the GWAS of birth weight or offspring birth weight ($p$-value $< 3.3 \times 10^{-9}$) are displayed in color, whilst the other 256 SNPs are shown with the gray open circles. Effect sizes and $p$-values represent conditional one-degree-of-freedom fetal and maternal tests calculated from the GWAS of birth weight and offspring birth weight using DINGO. The color of each point represents the $p$-value for the conditional maternal effect and the shape of each point the $p$-value for the conditional fetal effect. SNPs with large conditional effects are labeled with the name of the closest gene. Note the paucity of SNPs that show discordant maternal and fetal effects of similar magnitude. *It is possible that other SNPs at the same loci as these SNPs reached genome-wide significance in the GWAS of birth weight or offspring birth weight.

or offspring birth weight. Figure 5 shows the likely reason for this observation- most birth weight-associated SNPs that exhibited discordant maternal and fetal genetic effects have estimated maternal and fetal effect sizes of quite a different magnitude. Indeed, although there are some SNPs that show sizeable (discordant) maternal and fetal effects, these effects were so large that they were also picked up by one-degree-of-freedom tests in the separate GWASs of birth weight and offspring birth weight. In contrast, there were many SNPs that showed concordant maternal and fetal effects of similar magnitude, and in many of these cases the two-degree-of-freedom test was more powerful than separate GWASs (Fig. 5).

There were five and one SNPs that reached genome-wide significance in the GWAS of own birth weight and the GWAS of offspring birth weight, respectively ($p$-value $< 3.3 \times 10^{-9}$), but did not reach significance in the two-degree-of-freedom GWAS ($p$-value $< 6.6 \times 10^{-9}$); however, only two of these (both from the GWAS of own birth weight) involved loci not identified by the two-degree-of-freedom test. All these SNPs exhibited either predominantly fetal or maternal effects only (Supplementary Data 3).

Finally, the two-degree-of-freedom test outperformed the one-degree-of-freedom tests implemented in the popular multivariate GWAS package MTAG (Supplementary Fig. 6), which only identified 23 novel independent significant SNP associations with $p$-value $< 3.3 \times 10^{-9}$, all of which were identified in both the 2 degree of freedom $T_{2df}$ test and the one-degree-of-freedom meta-analyses described in the next section (Supplementary Data 4). In conditional analyses, all 23 of these SNPs had estimated maternal and fetal effects in the same direction.

## Empirical analyses of birth weight using one-degree-of-freedom meta-analyses

The one-degree-of-freedom meta-analytic strategy identified 369 independent SNPs at 226 loci that were genome-wide significant ($p$-value $< 3.3 \times 10^{-9}$) in either the meta-analysis of fetal effects, or the meta-analysis of maternal effects – an even greater number of loci than were identified using the two-degree-of-freedom strategy (Supplementary Data 5). These included 21 independent genome-wide significant SNPs at 21 loci that were not identified in the two-degree-of-freedom $T_{2df}$ tests (Fig. 6 and Supplementary Data 5), although all only narrowly missed out on significance in the two-degree-of-freedom $T_{2df}$ test, having a $p$-value smaller than $5 \times 10^{-8}$. Interestingly, eight independent SNPs (at eight loci) were significant in the two-degree-of-freedom $T_{2df}$ test but not in the meta-analysis of fetal effects, or the meta-analysis of maternal effects (Fig. 6 and Supplementary Data 2). Four of these SNPs exhibited significant ($p$-value $< 0.05$) discordant maternal and fetal effects in the conditional analyses– most notably rs7034200 at *GLIS3*, a known type 2 diabetes locus, that was highly significant in the two-degree-of-freedom test ($p$-value$_{2df} = 1.15 \times 10^{-12}$), but only exhibited relatively modest non-significant $p$-values in the maternal and fetal meta-analyses (minimum $p$-value$_{meta} = 7.39 \times 10^{-5}$), and likewise rs560887 at *G6PC2*, a known locus for fasting glucose which also exhibited radically different $p$-values across the two tests ($p$-value$_{2df} = 4.37 \times 10^{-20}$, minimum $p$-value$_{meta} = 6.25 \times 10^{-7}$). Five of these eight SNPs that were uniquely identified by the two-degree-of-freedom test were replicated in FinnGen (one-tailed $p$-value $< 0.05$) and seven out of eight displayed effects in the predicted direction (Supplementary Data 6).

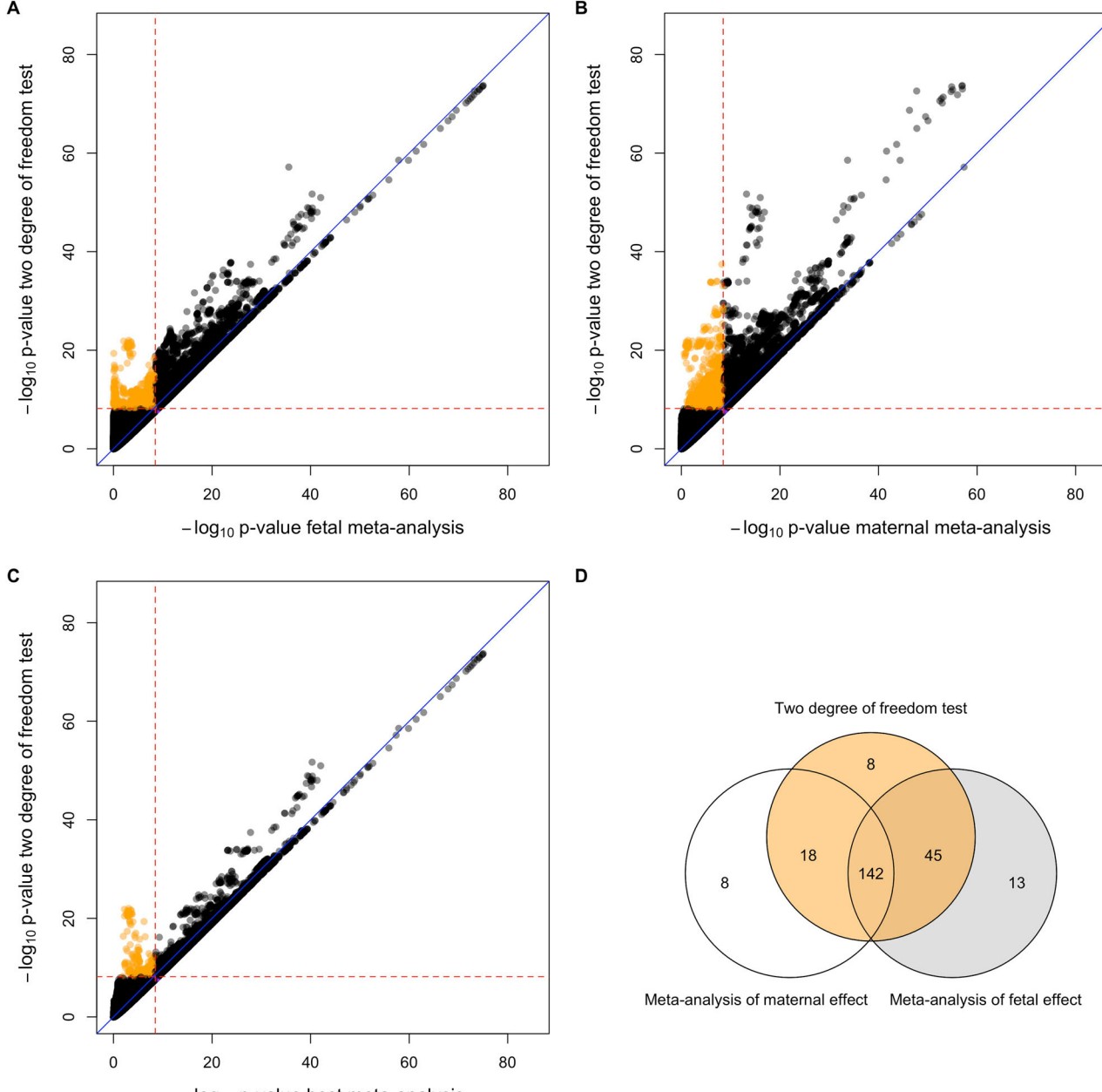

**Fig. 6 | *P*-value scatter plots and Venn diagram comparing one-degree-of-freedom meta-analysis tests versus the combined two-degree-of-freedom DINGO test for birth weight across 31,033,794 SNPs.** The -log$_{10}$*p*-value of the two-degree-of-freedom DINGO test calculated using the GWAS of fetal birth weight and the GWAS of maternal birth weight was compared against the -log$_{10}$*p*-value of SNPs from (**A**) the GWAS meta-analysis of fetal birth weight, (**B**) the GWAS meta-analysis of maternal birth weight, and (**C**) the stronger *p*-value from among **A** and **B**. Red dashed lines denote to genome-wide significant thresholds of $\alpha = 3.3 \times 10^{-9}$ for the one-degree-of-freedom maternal and fetal tests and $\alpha = 6.6 \times 10^{-9}$ for the two-degree-of-freedom DINGO test. Blue diagonal lines indicate x = y. Orange circles are SNPs that are only genome-wide significant in the two-degree-of-freedom test. There are a considerable number of SNPs that reach genome-wide significance in the GWAS meta-analysis but not in the two-degree-of-freedom test. These SNPs are all clustered in a very small region of Fig. 6a–c in the bottom right quadrant adjacent to the thresholds for genome-wide significance. **D** illustrates the overlap between genome-wide significant loci identified from one and two-degree-of-freedom tests.

We compared the results of our one-degree-of-freedom meta-analyses against the list of variants claimed as genome-wide significant in the most recent GWAS analysis of birth weight[5]. Our analyses identified an additional 68 SNPs at 62 novel loci compared to the deCODE paper (Supplementary Data 5). These included many SNPs previously associated with anthropometric traits (height, body mass index etc), blood pressure, bone mineral density, and type 2 diabetes at genome-wide significant levels (Supplementary Data 7). These are all phenotypes that are known to be causally or pleiotropically associated with birth weight,

increasing our confidence that our findings represent genuine associations. Follow-up in the FinnGen cohort revealed that 31 out of the 68 novel SNPs in Supplementary Data 5 had one-tailed *p*-value < 0.05 compared to only 3.4 expected under the null hypothesis of no association (Supplementary Data 8). Likewise, 60 of these 68 variants exhibited concordant directions of effect in the DINGO and FinnGen replication analyses (binomial sign test *p*-value < 0.0001). Together these results provide strong evidence that the novel loci identified in the DINGO analyses represent true associations. Finally, we also found

several new associations between SNPs and birth weight that could implicate new biological pathways in the etiology of the trait. These include an interesting association between rs11085720, an intronic variant in the DNA methylase transferase 1 (*DNMT1*) gene, and birth weight.

## Discussion

In this manuscript we compared the power of three strategies to detect loci in maternal and fetal GWASs of the same trait: (i) the traditional strategy of analysing maternal and fetal GWASs separately; (ii) a two-degree-of-freedom test which combines information from maternal and fetal GWASs; and (iii) a one-degree-of-freedom test where signals from maternal and fetal GWASs are meta-analysed together conditional on the estimated sample overlap. We found that which strategy performed best depended on the degree of sample overlap between maternal and fetal GWASs, the correlation between own and offspring phenotype, whether individual trait loci were simultaneously influenced by maternal and fetal effects, and if so, whether the effects were directionally concordant.

Our simulations showed that in general, when maternal and fetal GWASs contained only partially overlapping samples (e.g. as is the case for most published GWAS meta-analyses of perinatal traits to date), considerable gains in power could be achieved using the meta-analytic strategy or using the two-degree-of-freedom test compared to the traditional approach of analysing the scans separately. This makes sense intuitively since the traditional strategy of performing separate GWASs does not make use of the information contained across the scans. In these situations, for loci exhibiting concordant maternal and fetal effects (of similar magnitude) the two-degree-of-freedom and the meta-analytic strategy typically performed similarly, whereas for loci involving only maternal or only fetal effects, a simple meta-analysis was often most powerful. Our results imply that many existing GWASs for perinatal traits could benefit from a reanalysis using a two-degree-of-freedom test or meta-analytic strategy (a point we demonstrate in this manuscript with the empirical analysis of birth weight).

In contrast, when maternal and fetal GWASs involved completely overlapping samples (as is the case for cohorts where all individuals report their own and their offspring's phenotype), then which strategy was best depended on whether individual trait loci were simultaneously influenced by maternal and fetal effects, whether the effects were directionally concordant, and the background correlation between own and offspring phenotype. For loci that only exhibited effects through the fetal or maternal genome, then the traditional strategy of analysing maternal and fetal GWASs separately performed best. For loci that exhibited concordant maternal and fetal effects, the meta-analytic strategy was best (with the two-degree-of-freedom test a close second).

Interestingly, our simulations also showed that when a single locus exhibited maternal and fetal effects that operated in different directions, and were of similar magnitude, a two-degree-of-freedom test that modeled these effects often had vastly improved power to detect association compared to simple one-degree-of-freedom tests (regardless of the degree of sample overlap). This result is similar to what we have observed previously in tests of genetic association involving individual-level genotypes using a structural equation modeling framework[7,11]. Our results make sense intuitively as the two-degree-of-freedom test expends an extra degree of freedom, productively modeling both maternal and fetal effects, whereas the meta-analytic strategy combines effects in such a way that discordant effects may cancel out. Loci that exhibit maternal and fetal genetic effects in opposite directions are not uncommon in GWAS of perinatal traits related to growth[6,7]. The implication of our power analyses is that a proportion of these loci may be missed if only one degree-of-freedom tests are performed across the genome (i.e., either by meta-analysis or separate maternal and fetal GWASs).

In empirical analyses of birth weight, and consistent with our simulations, both the two-degree-of-freedom test and the meta-analytic strategy identified greater numbers of loci meeting genome-wide significance than analysing maternal and fetal GWASs separately. Both the meta-analytic strategy and the two-degree-of-freedom test also identified more genome-wide significant loci for birth weight than MTAG[13], a popular software package that is commonly used in the multivariate analysis of GWAS summary results statistics. This makes sense since the two-degree-of-freedom test (and the meta-analytic strategy) are derived by specifically modeling the correlation between maternal and fetal genotypes whereas other multivariate tests, e.g., MTAG[13], use models that are agnostic to the mother-offspring relationship and do not take advantage of this known source of information.

So which type of analysis is optimal in the case of birth weight and (potentially) other perinatal phenotypes, particularly in situations where there is only partial overlap between maternal and fetal GWASs? The meta-analytic strategy yielded considerably greater numbers of loci meeting genome-wide significance than the two-degree-of-freedom strategy (i.e. 226 vs 213 loci), suggesting that even for a trait like birth weight, a simple meta-analytic strategy may be optimal when maternal and fetal GWAs are only partially overlapping. However, whilst all loci that met genome-wide significance using the simple meta-analytic strategy were also significant or almost significant using DINGO (i.e. all loci would be flagged by researchers as significant or suggestive), many genome-wide significant loci that displayed directionally discordant effects in the two-degree-of-freedom DINGO test exhibited *p*-values that were far less extreme using the simple meta-analytic strategy, including some loci that may have been missed entirely (e.g. the *GLIS3* locus discussed below). The implication is that a two-degree-of-freedom test may be the preferable strategy in the case of phenotypes that are known to exhibit substantial numbers of loci with directionally discordant effects (e.g. perinatal growth-related traits like birth weight), whereas a simple meta-analytic strategy may be the superior strategy in the case of other traits (bearing in mind that such knowledge about likely underlying genetic architecture may not always be available from previous GWASs a priori). We have implemented both the meta-analytic tests and the two-degree-of-freedom test as part of the DINGO package in the publicly available web-based software CTG-VL[17].

We would like to highlight two findings from our empirical GWAS of birth weight that we think deserve further attention. The first is the *GLIS3* locus. As mentioned above, it is unlikely that this locus would have been flagged by researchers adopting a simple meta-analytic strategy, although interestingly the locus was identified previously in the deCODE paper using separate maternal and fetal GWASs (although not discussed explicitly by the authors). *GLIS3* encodes a member of the GLI-similar zinc finger protein family. The GLIS3 protein functions as both a repressor and activator of transcription that is involved in the development of pancreatic beta cells. Homozygous mutant *Glis3* mice develop neonatal diabetes due to insufficient pancreatic beta cells[18], and deletions of the gene in humans are associated with neonatal diabetes[19–21]. The A allele of the rs7034200 variant within the *GLIS3* gene (i.e. which in our study is associated with increased offspring birth weight when present in the maternal genome, but decreased birth weight when present in the fetal genome), has also been associated with increased risk of type 2 diabetes[22–27], higher fasting glucose levels[24,28,29] and impaired beta cell function[24,29–31]. The glucose-raising allele at this SNP is also strongly associated with impaired beta-cell function in non-diabetic adults[29,31] and in healthy children and adolescents[28]. We therefore hypothesize that diabetes-predisposing variants at this locus may increase levels of circulating glucose in pregnant mothers (i.e. which would tend to increase intrauterine growth and offspring birth weight), but simultaneously act to decrease offspring birth weight when the same alleles are transmitted to the

fetus (e.g. by impairing beta cell function and therefore fetal insulin secretion). Similar mechanisms are thought to underlie other diabetes-predisposing variants that show directionally inconsistent maternal and fetal genetic associations with birth weight[6,7,32].

The second result we would like to highlight is the genome-wide significant SNP rs11085720 in an intron in the *DNMT1* gene. Although this SNP was not significant in the deCODE meta-analysis, it was strongly significant in both the two-degree-of-freedom ($p$-value = $8.6 \times 10^{-13}$) and one-degree-of-freedom meta-analyses ($p$-value$_{min}$ = $2.3 \times 10^{-12}$). The gene *DNMT1* codes for the enzyme DNA methyltransferase 1 which is responsible for the addition of methyl groups to specific CpG structures in DNA. Methylation of CpG islands is associated with transcriptional silencing and knockout experiments suggest that this enzyme is responsible for the bulk of methylation in mouse cells and is essential for embryonic development[33]. In humans, rare variants in the *DNMT1* gene are associated with forms of cerebellar ataxia[34] and neuropathy[35,36]. Although we cannot be sure that *DNMT1* is indeed the causative gene at this locus, we highlight its potential involvement because of the considerable interest in the relationship between adverse environmental exposures during pregnancy, possible mediation via DNA methylation and long-term effects on disease risk due to putative intrauterine programming. In this respect, it is interesting to note that conditional analyses suggest that this locus involves both maternal and fetal effects. How methylation levels in mothers could in turn affect the birth weight of their offspring is unclear; however, it is interesting that studies of knock-out mice have shown that a lack of both maternal and zygotic *Dnmt1* results in complete demethylation of imprinted genes in (mouse) blastocysts[37].

Several simple extensions to/applications of our methods present themselves. First, although we have applied our methods to birth weight, there exist many other traits where maternal and offspring GWASs have been conducted including gestational duration[9,10,38] and placental weight[39] which involve only partially overlapping samples and could benefit from reanalysis using the framework that we have outlined in this manuscript. Second, there is increasing interest in and growing realization in the scientific community that indirect genetic effects contribute not only to perinatal traits like birth weight[1,6,7,40,41], but also to many later life socially patterned traits like educational attainment[14–16]. The implication is that substantial gains in power to detect loci underlying many of these traits might be achieved by combining parental and offspring GWASs either through a simple meta-analytic strategy, or by performing multi-degree of freedom tests across the genome to GWAS summary results data derived from parents and children. Third, although the development of DINGO was motivated by the analysis of traits that are affected by indirect maternal/paternal effects, there is no reason why analogous methods cannot also be applied to traits where e.g. the offspring's genome affects e.g. the mother's phenotype. For example, offspring genotype is thought to affect the risk of maternal preeclampsia so it would be possible in theory to combine a fetal GWAS of preeclampsia with a GWAS of maternal preeclampsia using a similar procedure for binary traits[42,43] (see below). Other phenotypes such as twinning and fertility/fecundity might also be interesting candidates for such analyses. Fourth, we note that our framework can be easily extended to a three degree of freedom test that incorporates summary results statistics from maternal, paternal, and fetal GWASs (see Supplementary Note). Whilst a three degree of freedom test would most likely be inefficient for the analysis of perinatal traits (i.e. where indirect paternal genetic effects are likely to be negligible and so the estimation of a paternal genetic effect parameter would be of little benefit whilst requiring an extra degree of freedom), such a test could potentially be useful for the analysis of traits like educational attainment which are known to involve all three sorts of effect[16]. Alternatively, employing a simple meta-analytic approach, but combining across maternal, paternal and offspring GWASs could also be advantageous (also see Supplementary

Note). Finally, it is possible that the power of our methods could be further improved by including additional information from adjacent SNPs in the same region as the index variant (i.e. a gene-based test performed on summary results data from maternal and fetal GWASs), although the development of such an approach is beyond the scope of the present manuscript.

There are several limitations to the methods proposed in this manuscript. First, the two-degree-of-freedom test and the meta-analytic strategy utilize LD score regression to provide an accurate estimate of effective sample overlap. Therefore, all factors discussed in the literature that may affect the accuracy/precision of the LD score intercept apply *mutatis mutandis* to the present methods. We recommend that the sample sizes of both maternal and fetal GWASs be large and as ancestrally homogenous as possible to accurately estimate sample overlap. Second, the formulae for the two-degree-of-freedom and meta-analytic tests that we have derived require that the coefficients from the maternal and fetal GWASs are from a linear regression analysis of quantitative traits. It is critically important that these different GWASs are analyzed on (or can be transformed to) the same scale. In the case of binary traits, several approaches exist for transforming logistic regression coefficients to a (normal) liability scale[44–47]. In theory it should be possible to transform these logistic coefficients (and their standard errors) to the liability scale and then apply the methods outlined in this manuscript to permit a similar analysis of binary traits. We expect that analogous to quantitative traits, such a method would improve the power to detect loci that simultaneously exert direct and indirect genetic effects and would accurately model their contribution to trait variation through maternal and fetal pathways. In contrast, existing methods for analysing binary data are geared towards increasing the power to detect direct genetic effects only through the utilization of self-report data from ungenotyped related individuals, or are appropriate for the analysis of individual level genotypes (i.e. rather than summary results data)[48–50]. Finally, our power simulations and empirical analyses used a Bonferroni correction to determine the genome-wide significance of the one-degree-of-freedom tests. Simulations we have conducted suggest that this correction is appropriate although very slightly conservative in the case of substantial sample overlap and phenotypic correlation (see Supplementary Note). Thus, our analyses may have slightly underestimated the true power of the one-degree-of-freedom tests under these conditions.

In conclusion, in this manuscript we have introduced two methods designed to increase the power of locus discovery in meta-analyses of summary results statistics from maternal and fetal GWASs. We have shown through a combination of analytical results, simulation, and empirical analysis that a simple one-degree-of-freedom meta-analytic strategy where signals from maternal and fetal GWASs are meta-analysed together conditional on the estimated sample overlap is likely to be a powerful strategy in the analysis of perinatal traits. Indeed, adopting this strategy increased the number of known birth weight loci by 62 and implicated several known and novel pathways in the etiology of the trait. We have also shown that a two-degree-of-freedom test may be a particularly powerful strategy for analysing traits where a substantial proportion of loci involve discordant maternal and fetal genetic effects of similar magnitude. The two-degree-of-freedom test and the one-degree-of-freedom meta-analytic procedure described in this manuscript are implemented in the Direct and INdirect effects analysis of Genetic lOci (DINGO) package, available as part of the CTG-VL software[17].

## Methods

### Formulation of the Two-degree-of-freedom Test
Simply regressing one's own phenotype on one's own genotype or offspring phenotype on maternal genotype will lead to inconsistent

and biased estimates of association in the presence of indirect maternal and direct fetal genetic effects at a locus (see Supplementary Note). We and others have shown previously how consistent and unbiased estimates of indirect maternal and direct fetal genetic effects at individual variants can be obtained by combining summary statistic linear regression coefficient estimates of SNP-trait associations from maternal and fetal GWASs[6,40], i.e.:

$$\hat{\beta}_f = \frac{4}{3}\hat{b}_f - \frac{2}{3}\hat{b}_m \tag{1}$$

$$\hat{\beta}_m = \frac{4}{3}\hat{b}_m - \frac{2}{3}\hat{b}_f \tag{2}$$

where $\hat{\beta}_f$ and $\hat{\beta}_m$ are estimates of the true population direct fetal ($\beta_f$) and indirect maternal genetic effects ($\beta_m$), and $\hat{b}_f$ and $\hat{b}_m$ are linear regression coefficients from a GWAS of own phenotype on own genotype, and offspring phenotype on maternal genotype, respectively. The standard error of these terms can be estimated using the below formulae:

$$\widehat{SE}(\hat{\beta}_f) = \sqrt{\frac{16}{9}\operatorname{var}(\hat{b}_f) + \frac{4}{9}\operatorname{var}(\hat{b}_m) - \frac{16}{9}\operatorname{cov}(\hat{b}_f, \hat{b}_m)} \tag{3}$$

$$\widehat{SE}(\hat{\beta}_m) = \sqrt{\frac{16}{9}\operatorname{var}(\hat{b}_m) + \frac{4}{9}\operatorname{var}(\hat{b}_f) - \frac{16}{9}\operatorname{cov}(\hat{b}_f, \hat{b}_m)} \tag{4}$$

where var and cov refer to the sampling variance and covariance of the terms respectively[6,40]. Note how the standard error of both terms is a function of the sampling covariance of $\hat{b}_f$ and $\hat{b}_m$. When GWAS of own and offspring traits are performed in separate unrelated families, this term can be assumed to be zero. However, this is rarely the case in practice. Wu et al. (2021)[40] show how this term is a function of the sample overlap and the phenotypic correlation between overlapping mothers and children, and can be estimated using the intercept from bivariate LD score regression[12]:

$$\widehat{\operatorname{cov}}(\hat{b}_f, \hat{b}_m) \approx \frac{N_S}{\sqrt{N_f N_m}}\rho\sqrt{\operatorname{var}(\hat{b}_f)\operatorname{var}(\hat{b}_m)} = \widehat{int} \times SE(\hat{b}_f)SE(\hat{b}_m) \tag{5}$$

where $N_f$ is the sample size of the GWAS of own genotype and own outcome (i.e. the fetal GWAS), $N_m$ is the sample size of the GWAS of maternal genotype and offspring phenotype (i.e. the maternal GWAS), $N_s$ serves to quantify the effective number of overlapping individuals across both GWASs (e.g. in situations where all mothers report their own and their offspring's birth weight the effective sample overlap would be 100%, whereas for genotyped mother-offspring pairs where only the offspring's phenotype is measured, the effective sample overlap would be 50%, since offspring and maternal genotypes are correlated by 0.5- these situations are illustrated in Supplementary Fig. 7), and $\widehat{int}$ is the estimated bivariate LD score regression intercept. The parameter $\rho$ refers to the correlation between own and offspring phenotype among the overlapping individuals.

In large samples, the hypothesis that the estimated maternal effect $\hat{\beta}_m$ (or estimated fetal effect $\hat{\beta}_f$) differs from zero can be evaluated for significance against the standard normal distribution, i.e.:

$$\frac{\hat{\beta}_f}{SE(\hat{\beta}_f)} \sim N(0,1) \tag{6}$$

$$\frac{\hat{\beta}_m}{SE(\hat{\beta}_m)} \sim N(0,1) \tag{7}$$

In this manuscript, we refer to these tests as conditional one-degree-of-freedom tests. These conditional one-degree-of-freedom tests have primary utility in locus characterization. That is, following the identification of the loci in GWASs, it is of interest to know whether the associated variant exerts its effect directly through the fetal genome, indirectly through the maternal genome, or some combination of both. However, we have shown that these conditional one-degree-of-freedom tests often lack statistical power[7,11] and typically do not produce stronger *p*-values than performing unconditional tests of SNP-trait association in mothers or children separately. In other words, whilst conditional one-degree-of-freedom tests are useful for locus characterization, in most situations they have limited utility for locus detection.

The joint sampling distribution of the estimated conditional maternal and fetal effects is bivariate normal in large samples i.e.:

$$\begin{pmatrix}\hat{\beta}_f \\ \hat{\beta}_m\end{pmatrix} \sim N\left(\begin{pmatrix}\beta_f \\ \beta_m\end{pmatrix}, \Sigma\right) \tag{8}$$

with

$$\Sigma = \begin{pmatrix}\operatorname{var}(\hat{\beta}_f) & \operatorname{cov}(\hat{\beta}_f, \hat{\beta}_m) \\ \operatorname{cov}(\hat{\beta}_f, \hat{\beta}_m) & \operatorname{var}(\hat{\beta}_m)\end{pmatrix} \tag{9}$$

To increase the power of locus identification in GWASs of own and offspring phenotypes, we propose $T_{2df}$, a two-degree-of-freedom test that considers the joint distribution of the two effect estimates. It follows from elementary statistical theory that under the null hypothesis ($H_0$) of no association between phenotype and maternal and fetal genotype (i.e. $\beta_f = \beta_m = 0$) the test statistic:

$$T_{2df} = \begin{pmatrix}\hat{\beta}_f & \hat{\beta}_m\end{pmatrix}\begin{pmatrix}\operatorname{var}(\hat{\beta}_f) & \operatorname{cov}(\hat{\beta}_f, \hat{\beta}_m) \\ \operatorname{cov}(\hat{\beta}_f, \hat{\beta}_m) & \operatorname{var}(\hat{\beta}_m)\end{pmatrix}^{-1}\begin{pmatrix}\hat{\beta}_f \\ \hat{\beta}_m\end{pmatrix} = \hat{\boldsymbol{\beta}}'\hat{\Sigma}^{-1}\hat{\boldsymbol{\beta}} \tag{10}$$

is asymptotically distributed as a $\chi^2$ statistic with two degrees of freedom (see the Supplementary Note for the derivation[51]). We verified using simulated data that $T_{2df}$ conforms with our expectations under the null (Supplementary Fig. 8 and Supplementary Data 9).

This manuscript primarily focuses on perinatal traits like birth weight that are known to be influenced by variants in both the maternal and offspring genomes. However, we also derive an analogous three degree of freedom test in the Supplementary Note that could be useful for identifying loci influencing phenotypes like educational attainment that are likely to be a function of direct genetic, indirect maternal, and indirect paternal effects.

### Formulation of One-degree-of-freedom Meta-analytic Tests

We also investigated the power of a simple procedure for meta-analysing the results of (potentially overlapping) maternal and fetal GWASs. We illustrate this procedure in the case of fetal genetic effects noting that an analogous test for maternal genetic effects can also be derived in a complementary fashion. We first assume that maternal genetic effects are not present at the locus. Since maternal and fetal genotypes are correlated by 0.5, we can obtain estimates of the fetal genetic effect from the maternal GWAS by simply doubling the estimated maternal effect regression coefficient (i.e. $2\hat{b}_m$), where $\hat{b}_m$ is the coefficient of the regression of offspring phenotype on maternal genotype. Likewise, the sampling variance of this estimate will be $4\operatorname{var}(\hat{b}_m)$. This estimate is then combined with estimates of the fetal genetic effect from the fetal GWAS by inverse variance weighted meta-analysis. The variance of this combined estimate is obtained by the usual inverse weighted variance formula plus an additional term due to

twice the weighted covariance between the estimates:

$$w_1 = \frac{1}{\text{var}(\hat{b}_f)} \quad (11)$$

$$w_2 = \frac{1}{4\text{var}(\hat{b}_m)} \quad (12)$$

$$\hat{\beta}_{f\,meta} = \frac{w_1 \hat{b}_f + 2w_2 \hat{b}_m}{w_1 + w_2} \quad (13)$$

$$\text{SE}(\hat{\beta}_{f\_meta}) = \sqrt{\left(\frac{w_1}{w_1+w_2}\right)^2 \text{var}(\hat{b}_f) + 4\left(\frac{w_2}{w_1+w_2}\right)^2 \text{var}(\hat{b}_m) + 4\text{cov}(\hat{b}_f, \hat{b}_m)\frac{w_1 w_2}{(w_1+w_2)^2}} \quad (14)$$

where $\hat{b}_f$ is the coefficient from the regression of own phenotype on own genotype and $\hat{b}_m$ is the coefficient from the regression of offspring phenotype on maternal genotype, and $\hat{\beta}_{f\_meta}$ is the inverse variance weighted estimate of the fetal effect across both these scans. The effective sample overlap between the scans is estimated using bivariate LD score regression and the covariance between the regression coefficients estimated from this quantity:

$$\widehat{\text{cov}}\left(\hat{b}_f, \hat{b}_m\right) \approx \frac{N_S}{\sqrt{N_f N_m}} \rho \sqrt{\text{var}(\hat{b}_f)\text{var}(\hat{b}_m)} = \widehat{int} \times \text{SE}(\hat{b}_f)\text{SE}(\hat{b}_m) \quad (15)$$

In large samples, under the null hypothesis of no association between SNP and trait (i.e. fetal or maternally mediated), the regression coefficients over their standard errors are distributed as standard normal:

$$\frac{\hat{\beta}_{f\_meta}}{\text{SE}(\hat{\beta}_{f\_meta})} \sim N(0, 1) \quad (16)$$

We refer to this test/procedure as the one-degree-of-freedom meta-analytic strategy. We note that when both maternal and fetal effects are present at a locus, simply meta-analysing the data as described above will result in biased estimates of maternal and fetal genetic effects. However, since our interest is on locus discovery (i.e. regardless of whether a genetic effect is mediated by the fetal or maternal genome), we argue that this bias can be ignored, as the type I error for both tests should be well-calibrated under the null (i.e. when neither maternal nor fetal effects contribute to trait variation at the locus). Indeed, if desired, unbiased estimates of maternal and fetal genetic effects can be obtained post-hoc using the two-degree-of-freedom procedure described in the previous section. In Supplementary Data 9, we show that the type I error rate for the meta-analytic tests is well calibrated.

With a couple of notable exceptions (e.g. the Norwegian MoBa cohort), the majority of the world's birth cohorts contain only limited genotype information on the children's fathers. Nevertheless, in some situations, it may be useful to include information from a GWAS of offspring phenotype regressed on father's genotype. This could be useful when paternal effects are suspected to contribute to offspring trait variation (e.g. in the case of educational attainment[52]), or (in their absence) when fathers provide information on their offspring in the absence of offspring genotype information (i.e. in which case the fathers are providing information indirectly on offspring mediated effects). In the Supplementary Note we derive analogous formulae for meta-analyses when paternal, maternal, and fetal GWASs are available.

## Power

We performed extensive data simulations to investigate the power of the three locus detection strategies described in this manuscript i.e. (i) the usual strategy of separately analysing maternal and fetal GWASs with one-degree-of-freedom tests; ii) two-degree-of-freedom $T_{2df}$ tests applied to the maternal and fetal GWASs concurrently; and (iii) meta-analysing maternal and fetal GWASs to obtain estimates of the maternal and fetal effect using one-degree-of-freedom tests. We simulated bivariate normally distributed quantitative traits of unit variance for own and offspring phenotype that were influenced by a single additive locus ($N = 1000$ simulations). We varied fetal effect size ($\beta_f$ = -0.03, -0.01, 0, 0.01, or 0.03), maternal effect size ($\beta_m = 0$ or 0.03), residual correlation between maternal and fetal phenotype (three conditions: (i) $\rho = 0$; (ii) $\rho = 0.25$; (iii) $\rho = 0.5$), sample size of each GWAS (five conditions: N = 10000, 20000, 30000, 40000, or 50000), relative sample size (two conditions: (i) equal sized maternal and fetal GWASs; (ii) maternal GWAS half the size of fetal GWAS) and extent of sample overlap (three conditions: (i) zero overlap; (ii) 50% of individuals in the maternal GWAS in the fetal GWAS; (iii) all of the mothers in the maternal GWAS are also in the fetal GWAS).

Since the locus detection strategies involving one-degree-of-freedom tests entail performing double the number of statistical tests relative to the two-degree-of-freedom test (i.e. performing genome-wide tests for a fetal effect, and genome-wide tests for a maternal effect), we evaluated power for the one-degree-of-freedom tests against a genome-wide Bonferroni corrected alpha level of $\alpha/2 = 6.6 \times 10^{-9}/2 = 3.3 \times 10^{-9}$ (for a justification of why a Bonferroni correction is reasonable, see the Supplementary Note). However, since our focus is on locus discovery, and so technically we are agnostic as to whether evidence for association comes from the maternal or fetal meta-analysis, we evaluated power according to whether *either* the maternal *or* the fetal test was significant for a given replicate. Note that for our data simulations, we adopted a slightly more stringent significance level than the typical $\alpha = 5 \times 10^{-8}$ threshold employed in most GWASs because of the increased number of low-frequency variants contained within modern imputation panels (see Morris et al. (2019) for further justification of this threshold)[53].

In the Supplementary Note we derive analytical expressions for the non-centrality parameter (and hence power) of all the statistical tests described in this manuscript and confirm that the asymptotic power implied by the expressions matches the results of our simulations closely. We encode several of our asymptotic formulae into an online web calculator that allows users to approximate power in their own GWASs (https://evansgroup.di.uq.edu.au/DINGO).

## Empirical application

We applied our strategies to birth weight GWAS summary statistics from the most recent GWAS of birth weight[5] which contains meta-analysed data from deCODE, the EGG Consortium, and the UK Biobank. The GWAS of own birth weight included 423,683 individuals of European ancestry and the GWAS of offspring birth weight included 270,002 women of European ancestry as defined by the authors of the study (https://www.decode.com/summarydata/). Given that the standard errors were not provided in the GWAS summary statistics, we derived them using p-values and beta effect estimates and the qnorm function in R, i.e., standard error = abs(beta/qnorm(p-value/2)). Among 33,409,268 single nucleotide polymorphisms (SNPs) from the GWAS of own birth weight and 33,453,671 SNPs from the GWAS of offspring birth weight, 1,276,496 and 1,220,850 SNPs were removed, respectively, due to p-value = 1 or beta = 0 because in either of these conditions standard errors cannot be derived. For the remaining SNPs, 31,033,794 were available in both GWASs and were included in the two-degree-of-freedom $T_{2df}$ test and in the meta-analyses of the estimated fetal and maternal effects. We performed

clumping analyses for SNPs within a 500-kb window size using $r^2 = 0.05$, and $\alpha = 6.6 \times 10^{-9}$, and the 1000 Genome Phase 3 EUR reference panel using the online platform FUMA[54]. We compared genome-wide significant loci from the two-degree-of-freedom test against those from the published GWAS that used a one-degree-of-freedom test (evaluated at $\alpha = 3.3 \times 10^{-9}$) and with loci identified in the meta-analyses of maternal/fetal effects (evaluated at $\alpha = 3.3 \times 10^{-9}$). For genome-wide significant SNPs, we then applied conditional one-degree-of-freedom tests (evaluated at $\alpha = 0.05$) to test for whether effects were likely to be mediated through the maternal and/or fetal genomes. Finally, for lead variants at loci not reported as genome-wide significant in the most recent deCODE meta-analysis of birth weight[5], we searched the GWASs Catalog and variants in strong linkage disequilibrium ($r^2 > 0.9$) for reported trait associations using the FUMA software package[54]. This study was approved by the Human Research Ethics Committee at the University of Queensland (approval number: 2019002705)

## Comparison with other multivariate software

$T_{2df}$ differs from other commonly used multi-trait association analyses that do not precisely model the correlation between the parents and their offspring. We therefore compared the performance of $T_{2df}$ against MTAG (evaluated at $\alpha = 3.3 \times 10^{-9}$), a commonly used tool for boosting statistical power to detect loci underlying correlated traits using summary results statistics[13]. We applied MTAG to the same summary results statistics of fetal and maternal birth weight as described above and compared the results against those from $T_{2df}$.

## Replication analyses

To investigate whether novel loci identified by our methods were likely to represent genuine loci, we performed replication analyses in 55,656 individuals with data on their own birth weight from the FinnGen cohort[55]. Further details regarding FinnGen can be found in the Supplementary Note.

## Reporting summary

Further information on research design is available in the Nature Portfolio Reporting Summary linked to this article.

## Data availability

GWAS summary results statistics of birth weight published by deCODE are available at https://www.decode.com/summarydata and can be downloaded here https://download.decode.is/form/2021/Birthweight2021.gz and here https://download.decode.is/form/2021/Birthweight_offspring_mothers2021.gz. GWAS summary results statistics from the DINGO analysis of birth weight generated in this study have been deposited in the UQ Research Data Manager at https://doi.org/10.48610/9f69854. The UK Biobank data (https://www.ukbiobank.ac.uk/) were accessed with application ID 53461. The FinnGen data are restricted and can be requested at (https://www.finngen.fi/en).

## Code availability

The source code of the DINGO software is available at https://github.com/danielldhwang/DINGO. R code used to perform empirical analyses of birth weight is provided in the Supplementary Note.

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

## Acknowledgements

This work was supported by Australian National Health and Medical Research Council (NHMRC) grants (GNT1183074, GNT1157714). D.M.E. is supported by an NHMRC Leadership Fellowship (2017942). L.D.H. is supported by an Australian Research Council Discovery Early Career Researcher Award (DE240100014). N.M.W. is supported by an NHMRC Emerging Leadership Fellowship (2008723). G.H.M. is the recipient of an Australian Research Council Discovery Early Career Award (DE220101226) funded by the Australian Government and supported by the Research Council of Norway (Project grant: 325640). L.Y. is supported by an Australian Research Council Future Fellowship (FT220100069). R.M.F. and R.N.B. are supported by a Wellcome Senior Research Fellowship (WT220390). R.M.F. is also supported by a grant from the Eunice Kennedy Shriver National Institute Of Child Health & Human Development of the National Institutes of Health under Award Number R01HD101669. This research was funded in part by the Wellcome Trust (WT220390). Human genotype and phenotype data from the UKB on which the results of simulations were based were accessed with accession ID 53641. We would like to thank Geng Wang for assistance in proofreading the manuscript. We would like to acknowledge the participants and investigators of the FinnGen study.

## Author contributions

L.D.H. led the data analyses, created figures, wrote the manuscript, and developed the software. G.C.P. contributed to data analyses and developed the software. L.Y. contributed to data analyses. J.Z. contributed to data analyses. J.T. contributed to data analyses. M.A. contributed to data analyses. R.N.B. contributed to data analyses. R.M.F. contributed to data analyses. G.H.M. contributed to data analyses and created the web power calculator. N.M.W. contributed to data analyses. D.M.E. conceived the study, wrote the manuscript, and supervised the entire project. All authors reviewed and approved the manuscript.

## Competing interests

G.C.P. is an employee of Gilead Sciences, Inc. and may hold stock of the company. All other authors declared no competing interests.
