## [Transparent Peer Review file · Nature Communications]

DINGO: increasing the power of locus discovery in maternal and fetal genome-wide association studies of perinatal traits

Corresponding Author: Professor David Evans

Version 0:

Reviewer comments:

Reviewer #1

(Remarks to the Author)

The authors propose a novel approach for detecting loci in maternal and fetal GWAS of the same trait that conducts a computationally efficient two degrees of freedom test and investigate its performance by comparing it to a simple one degree of freedom test with signals from maternal and fetal GWAS being meta-analyzed conditional on the estimated sample overlap, as well as to the traditional approach of separately analyzing maternal and fetal GWAS. The power of the three approaches is investigated through analytical formulae, data simulation, and real data analyses, and a web tool is provided to conduct asymptotic power calculations for the various approaches. Some interesting results from an empirical analysis of birth weight were discussed. While the method and results are intriguing, the writing can be improved (you can spot many grammar issues by a quick scan).

Comments

1. In the simulation study, the author “simulated bivariate normally distributed quantitative traits of unit variance for own and offspring phenotype that were influenced by a single additive locus” and assumed a sample size of 1000 to 10000 for each GWAS. While the simulation results under these simulation settings can be informative, the author should consider adding a simulation study under more realistic settings that shows what relative performance of the 3 tests looks like under different GWAS sample sizes (e.g., 5000 – 30,000, as seen in real GWAS) and genetic architecture of the trait (causal SNP proportion, heritability, concordant/discordant maternal and fetal effects of similar or different magnitudes, etc.). In the simulation results the author should also report and briefly comment the type I error control of the 3 tests, e.g., are the typical GWAS sample sizes (e.g., 5000, 50,000, etc.) sufficient to have well controlled type I error rates.
2. Lines 267-268, “We confirm that our formulae for asymptotic power of the two degree of freedom test derived in the Supplementary Note gave similar results to those obtained using simulated data (results not shown)”. The statement is too vague. Some figures showing the approximation under different sample sizes would be helpful, unless there is a reason to not show the results (e.g., too difficult to summarize the results).
3. I’m curious if including all SNPs by accounting for LD will increase the power of locus discovery in this problem setting. And if so, can the proposed test be extended to incorporate genome-wide SNPs instead of only testing for a set of relatively independent SNPs?
4. One way to evaluate the performance of the different tests is through some follow-up analyses, e.g., constructing a polygenic score (PGS) using the identified SNPs from each method and calculating its R² for polygenic prediction, on a separate validation dataset. If an individual-level validation dataset is available, it would be great to add some follow-up analyses to further confirm that the additional loci the proposed one degree and two degree of freedom tests identified truly have signals.
5. In the abstract (also later on page 3, line 65), the author states that “Both the two degree of freedom (213 loci) and meta-analytic approach (226 loci) ‘dramatically’ increase the number of robustly associated genetic loci for birth weight relative to separately analysing the scans (183 loci).” It may be prudent to avoid characterizing the increase as “dramatic”. A less strong statement might be more appropriate here.

6. In the supplementary notes, line 4: “f_i” and “m_i” may be more appropriate to be used as the the subscript of x instead of “f_j” and “m_j”.

7. Lines 113 – 115, “However, we have shown that these conditional one degree of freedom tests often lack statistical power and typically do not produce stronger p-values than performing (potentially biased) unconditional tests of SNP-trait association in mothers or children separately.” The authors should point out that both methods have well controlled type I error rate. Otherwise, the readers may think that since the unconditional, separate tests can lead to biased results, it makes sense that the conditional one degree of freedom tests may give larger p-values, which does not necessarily mean it lacks statistical power but rather means it has better type I error control. The authors should consider rephrasing the sentence to make it clearer.

8. Supplementary Figure 1 legend, the authors should also mention the one degree of freedom test statistic and explain the rationale of showing the approximation of both the one degree of freedom test and the two degree of freedom test statistics in the figure.

Reviewer #2

(Remarks to the Author)

In this manuscript, the authors propose a software package to increase the power to detect genomic locus with either maternal or fetal effect on the offspring’s quantitative phenotypes. Conventionally, maternal and fetal effects are tested through separate GWASs, and the estimates are likely biased due to the correlation between maternal and fetal genomes. The authors attempt to correct the biases and further conduct either a 2-df test or a 1-df meta-analysis. The tests are based on summary statistics. Both simulations and real data applications are presented. I found the major strength to be the modelling strategy to correct biases in effect estimates. I also have some doubts about its potential to improve gene discovery. Here are my comments:

1. Some clarifications are needed regarding the concepts of “direct effect” and “indirect effect” discussed in the manuscript. As I understand, these terms are commonly used to describe a causal pathway. The indirect pathway is the effect of an exposure on the outcome that works through a mediator. The introduction seems to suggest it is the question of interest, but I do not think it is correct in the context of this study. If a maternal gene functions through the mediating effect of a fetal gene, it is no longer a maternal effect but a fetal effect. It is conceivable that a maternal effect will be mediated by intrauterine environment to influence phenotype. However, fetal gene is not involved in the pathway, and it seems to me that the distinction of “direct” or “indirect” does not matter with respect to fetal effect. In fact, following similar logic, one may claim that fetal effect is “indirect” and mediated by molecular and physiological changes within the offspring. The presumed true model in the supplementary note also only describes the maternal and fetal effects. It seems that the authors are using “direct” and “indirect” to describe fetal and maternal effect, respectively. In my opinion, it causes a lot of confusions. The challenge appears to be that the estimated effects from fetal GWAS and maternal GWAS are likely be biased due to the confounding effects of one another. The proposed strategies showed merits related to this challenge, but the topic would be “maternal and fetal effect analysis of genetic loci” rather than “direct and indirect effect analysis of genetic loci”.

2. I also have some doubts about the goal of this study as presented. As the authors stated, the goal of the study is for gene discovery. Therefore, the effect estimates might be less critical as long as the separate GWAS do not miss genes with either maternal or fetal effects. I wonder if separate GWAS without additional multiple testing adjustment (2 times) will identify the majority of the findings by the proposed strategies. There are also strategies to integrate testing p-values that are corrected so that multiple-testing adjustment is not needed.

3. I think an evaluation of type 1 error is necessary. The supplementary figure 1 shows that the overall distribution of test statistics is not significantly different from the theoretical one. However, in large-scale genetic studies, what matters most is to have accurate type I errors at the tail of distribution (e.g., $<5 \times 10^{-8}$).

4. At the end of abstract, the author concluded that “a simple meta-analytic strategy is likely to perform best, particularly in situations where maternal and fetal GWAS only partially overlap.” I am not sure this is reflecting the simulation and application results in Fig 1 and Fig 2. Fig 1 is difficult to read (maybe better color and legend can help). However, it is difficult for me to conclude from this figure that meta-analytic has meaningful improvement over 2-df test. It appears that the meta-analytic may have slightly improved power, which does not seem significant, while may suffer substantial power loss when the effects are bi-directional. Figure 2 also seems to suggest 2-df has improved power with smaller p-values.

5. Would it be reasonable to conduct 1-df meta-analytics for the coefficients estimated from separate GWAS similar to the proposed strategy? Again, if the goal is truly gene discovery, would it have similar performance though the effect estimates are biased?

6. I am confused with the descriptions of sample overlaps, information overlap, independence, etc. It seems that the authors

use 100% sample overlap for studies with mother-offspring pairs. I think I understand what you are trying to say, but it does not seem correct to describe it as 100% sample overlap. There are overlapping information also independent information.

7. The method, as currently proposed, is for quantitative phenotypes, which needs to be reflected in the title and abstract.

8. The authors discussed the potential extension to 3-df test with paternal effect. It is hard to believe that paternal effect will function without transmitting the risk allele (fetal effect). Please provide references if such examples exist.

9. Minor comments. There are other confusing descriptions and errors impact the understanding.

- “own genotype” and “own phenotype” are used to refer either maternal or fetal data at different places is confusing. I suggest being consistent with maternal GWAS and fetal GWAS.
- Line 48, should read “due to multiple testing”.
- “two-degree” instead of “two degree”
- “information shared across the individual GWAS” and “information contained across both GWAS” at different places. Also, the use of “GWAS” as both single and plural forms.
- Line 50, the sentence is problematic.
- Explain abbreviations, such as MTAG.

Version 1:

Reviewer comments:

Reviewer #1

(Remarks to the Author)

1. Supplementary Figures 4 and 6: I can only see the blue lines but not the other two? Maybe the authors can recreate the figures to make the results of the other two methods visible?
2. It's hard to see which ones are type one error rates in Supplementary Table 2. I'm also confused about what is the level of type I error control here. It seems like when $\beta_m = 0$ and $\beta_f = 0$, all p-values are 0?

Reviewer #2

(Remarks to the Author)

The authors have responded to my comments.

Version 2:

Reviewer comments:

Reviewer #1

(Remarks to the Author)

(Remarks on code availability):

The code looks fine. However, I could not check the reproducibility of the results because no data was provided.

Version 3:

Reviewer comments:

Reviewer #1

(Remarks to the Author)

The authors have addressed my concerns and I have no further comment.

REVIEWER COMMENTS

Reviewer #1 (Remarks to the Author):

The authors propose a novel approach for detecting loci in maternal and fetal GWAS of the same trait that conducts a computationally efficient two degrees of freedom test and investigate its performance by comparing it to a simple one degree of freedom test with signals from maternal and fetal GWAS being meta-analyzed conditional on the estimated sample overlap, as well as to the traditional approach of separately analyzing maternal and fetal GWAS. The power of the three approaches is investigated through analytical formulae, data simulation, and real data analyses, and a web tool is provided to conduct asymptotic power calculations for the various approaches. Some interesting results from an empirical analysis of birth weight were discussed. While the method and results are intriguing, the writing can be improved (you can spot many grammar issues by a quick scan).

We have run the manuscript through Grammarly and Microsoft 365 before resubmission. If the reviewer has specific instances of poor grammar that they would like to highlight and suggestions for correction, we will gladly change these upon revision.

Comments

1. In the simulation study, the author “simulated bivariate normally distributed quantitative traits of unit variance for own and offspring phenotype that were influenced by a single additive locus” and assumed a sample size of 1000 to 10000 for each GWAS. While the simulation results under these simulation settings can be informative, the author should consider adding a simulation study under more realistic settings that shows what relative performance of the 3 tests looks like under different GWAS sample sizes (e.g., 5000 – 30,000, as seen in real GWAS) and genetic architecture of the trait (causal SNP proportion, heritability, concordant/discordant maternal and fetal effects of similar or different magnitudes, etc.). In the simulation results the author should also report and briefly comment the type I error control of the 3 tests, e.g., are the typical GWAS sample sizes (e.g., 5000, 50,000, etc.) sufficient to have well controlled type I error rates.

We agree that it would be useful to examine more simulations for concordant/discordant maternal and fetal effects of similar and/or different magnitudes, sample and effect sizes that are more typical of current GWAS, and more degrees of sample overlap, rather than just the extremes. We have included the following conditions in the methods section of the manuscript:

“We varied fetal effect size ($\beta_f = -0.03, -0.01, 0, 0.01, \text{ or } 0.03$), maternal effect size ($\beta_m = 0$ or 0.03), residual correlation between maternal and fetal phenotype (three conditions: (i) $\rho = 0$;

(ii) $\rho = 0.25$; (iii) $\rho = 0.5$), sample size of each GWAS (5 conditions: $N = 10000, 20000, 30000, 40000, \text{ or } 50000$), relative sample size (two conditions: (i) equal sized maternal and fetal GWASs; (ii) maternal GWAS half the size of fetal GWAS) and extent of sample overlap (three conditions: (i) zero overlap; (ii) 50% of individuals in the maternal GWAS in the fetal GWAS; (iii) all of the mothers in the maternal GWAS in the fetal GWAS).”

We have incorporated the results of these simulations into the manuscript (Figure 1, Supplementary Table 2, Supplementary Figures 3 - 7). Type I error rates are now included in Supplementary Table 2. We can confirm that type 1 error rates are well calibrated for all conditions we examined. In addition, we investigated type I error rate using thinned results from permuted UK Biobank replicates (Supplementary Table 1). Our methods display good control for type I error rate across all the thresholds we examined for all methods.

Since all the statistical tests considered in this manuscript are single locus tests that do not model heritability and/or causal SNP proportion, these parameters are irrelevant in terms of power. This can be seen by looking at the formulae for the non-centrality parameters in the Supplementary Note which do not depend on these quantities.

2. Lines 267-268, “We confirm that our formulae for asymptotic power of the two degree of freedom test derived in the Supplementary Note gave similar results to those obtained using simulated data (results not shown)”. The statement is too vague. Some figures showing the approximation under different sample sizes would be helpful, unless there is a reason to not show the results (e.g., too difficult to summarize the results).

We have now included the asymptotic power estimated by our online power calculator in Supplementary Table 2 across all conditions. Supplementary Table 2 shows that there is excellent agreement between the asymptotic results and our simulations in all cases. We have amended our results section to reflect this fact.

3. I’m curious if including all SNPs by accounting for LD will increase the power of locus discovery in this problem setting. And if so, can the proposed test be extended to incorporate genome-wide SNPs instead of only testing for a set of relatively independent SNPs?

It is possible that including additional signals from SNPs in LD across a region might improve the statistical power of our tests even further (e.g. as part of a “gene-based” test, c.f. Li et al 2023 AJHG). However, the development of such tests is beyond the scope of an already very long manuscript. We have included the following text in the Discussion that raises this possibility:

“Finally, it is possible that the power of our methods could be further improved by including additional information from adjacent SNPs in the same region as the index variant (i.e. a gene-based test performed on summary results data from maternal and fetal GWAS),

although the development of such an approach is beyond the scope of the present manuscript.”

Note that our group has previously developed statistical methods and software to estimate the proportion of phenotypic variance attributable to maternal and fetal effects (and their covariance) using individual level genome-wide SNP data (c.f. Eaves et al. 2014 Behav Genet; Qiao et al. 2020 Behav Genet; Eilertsen et al. 2021 Behav Genet), and genome-wide summary results data (Moen et al. 2023 Behav Genet) as opposed to single variants which are the focus of this manuscript.

4. One way to evaluate the performance of the different tests is through some follow-up analyses, e.g., constructing a polygenic score (PGS) using the identified SNPs from each method and calculating its R² for polygenic prediction, on a separate validation dataset. If an individual-level validation dataset is available, it would be great to add some follow-up analyses to further confirm that the additional loci the proposed one degree and two degree of freedom tests identified truly have signals.

We have now included replication analyses from the FINNGEN cohort (i.e. who were not part of the deCODE or Warrington et al. birthweight meta-analyses) in our paper. Despite limited power to detect association given the considerable difference in sample sizes between discovery and replication sets (i.e. >400,000 vs ~56,000 individuals), we found that 31 of the 68 novel SNPs presented in Table 1 had one tailed $p < 0.05$ in the FINNGEN replication analyses compared to only 3.4 expected under the null hypothesis of no association (now presented in Supplementary Table 9). 60 of these 68 variants exhibited concordant directions of effect in the DINGO and FINNGEN replication analyses (binomial sign test: $p < 0.0001$). Likewise, five of the eight SNPs that were uniquely identified by the two degree of freedom test were replicated in FINNGEN (one tailed $p < 0.05$) (now presented in Supplementary Table 7). These results provide strong evidence that the novel DINGO results represent genuine loci. The details of these replication analyses have now been added to the manuscript.

5. In the abstract (also later on page 3, line 65), the author states that “Both the two degree of freedom (213 loci) and meta-analytic approach (226 loci) ‘dramatically’ increase the number of robustly associated genetic loci for birth weight relative to separately analysing the scans (183 loci).” It may be prudent to avoid characterizing the increase as "dramatic". A less strong statement might be more appropriate here.

We have removed the word “dramatic” from the sentence.

6. In the supplementary notes, line 4: “f_i” and “m_i” may be more appropriate to be used as the subscript of x instead of “f_i” and “m_i”.

The relevant quantities are already expressed as subscripts without separation by underscores in our version of the manuscript. It is possible that the software used by the journal website may have added the underscores during the upload process.

7. Lines 113 – 115, “However, we have shown that these conditional one degree of freedom tests often lack statistical power and typically do not produce stronger p-values than performing (potentially biased) unconditional tests of SNP-trait association in mothers or children separately.” The authors should point out that both methods have well controlled type I error rate. Otherwise, the readers may think that since the unconditional, separate tests can lead to biased results, it makes sense that the conditional one degree of freedom tests may give larger p-values, which does not necessarily mean it lacks statistical power but rather means it has better type I error control. The authors should consider rephrasing the sentence to make it clearer.

We have removed the words “potentially biased” from the sentence.

8. Supplementary Figure 1 legend, the authors should also mention the one degree of freedom test statistic and explain the rationale of showing the approximation of both the one degree of freedom test and the two degree of freedom test statistics in the figure.

We have removed the chi-square one-degree-of-freedom distribution from the Supplementary Figure since it is unnecessary.

Reviewer #2 (Remarks to the Author):

In this manuscript, the authors propose a software package to increase the power to detect genomic locus with either maternal or fetal effect on the offspring’s quantitative phenotypes. Conventionally, maternal and fetal effects are tested through separate GWASs, and the estimates are likely biased due to the correlation between maternal and fetal genomes. The authors attempt to correct the biases and further conduct either a 2-df test or a 1-df meta-analysis. The tests are based on summary statistics. Both simulations and real data applications are presented. I found the major strength to be the modelling strategy to correct biases in effect estimates. I also have some doubts about its potential to improve gene discovery. Here are my comments:

1. Some clarifications are needed regarding the concepts of “direct effect” and “indirect effect” discussed in the manuscript. As I understand, these terms are commonly used to describe a causal pathway. The indirect pathway is the effect of an exposure on the outcome that works through a mediator. The introduction seems to suggest it is the question of interest, but I do not think it is correct in the context of this study. If a maternal gene functions through the mediating effect of a fetal gene, it is no longer a maternal effect but a fetal effect. It is conceivable that a maternal effect will be mediated by intrauterine environment to influence phenotype. However, fetal gene is not

involved in the pathway, and it seems to me that the distinction of “direct” or “indirect” does not matter with respect to fetal effect. In fact, following similar logic, one may claim that fetal effect is “indirect” and mediated by molecular and physiological changes within the offspring. The presumed true model in the supplementary note also only describes the maternal and fetal effects. It seems that the authors are using “direct” and “indirect” to describe fetal and maternal effect, respectively. In my opinion, it causes a lot of confusions. The challenge appears to be that the estimated effects from fetal GWAS and maternal GWAS are likely be biased due to the confounding effects of one another. The proposed strategies showed merits related to this challenge, but the topic would be “maternal and fetal effect analysis of genetic loci” rather than “direct and indirect effect analysis of genetic loci”.

Referring to the effect of relatives’ genotypes on one’s own phenotype as “indirect genetic effects” and referring to the effect of one’s own genotype on one’s own phenotype as “direct genetic effects” is standard practice in the genetics literature - c.f. recent papers in *Nature Genetics* (Howe et al. 2022; Young et al. 2022). We adopt a similar nomenclature in this manuscript.

We feel that it is important to keep the “direct” and “indirect” nomenclature because our methods have broader application than merely modelling “maternal” and “fetal” effects (i.e. our methods also apply to paternal genetic effects- another form of indirect genetic effect, and also effects from the offspring genome on the offspring phenotype that develop postnatally after the fetal period).

However, in order to make the direct/indirect distinction clearer we have modified the introduction of the manuscript to read:

“Genetic variants showing genome-wide significant associations in either scan are then followed up using conditional association analyses (or transmitted and non-transmitted haplotype analyses) to investigate whether their effects are due to the fetal genotype, the maternal genotype, or some combination of both (NB. In this manuscript we refer to the effect of the maternal genotype on the offspring phenotype as an indirect maternal genetic effect, since the effect is mediated indirectly through the intrauterine environment. In contrast, we refer to the effect of an individual’s own genotype on their own phenotype (here birthweight), as a direct fetal genetic effect).”

2. I also have some doubts about the goal of this study as presented. As the authors stated, the goal of the study is for gene discovery. Therefore, the effect estimates might be less critical as long as the separate GWAS do not miss genes with either maternal or fetal effects. I wonder if separate GWAS without additional multiple testing adjustment (2 times) will identify the majority of the findings by the proposed strategies. There are also strategies to integrate testing p-values that are corrected so that multiple-testing adjustment is not needed.

Evaluating the significance of separate maternal and fetal GWAS without multiple testing adjustment is not an appropriate strategy, since by definition, it will increase the number of

false positive loci. Accordingly, it is not fair to compare separate GWAS without multiple testing adjustment with the other methods examined in this manuscript that do control type I error rates appropriately (e.g. the two-degree-of-freedom test). Note that we have already investigated the degree to which a Bonferroni correction of the findings may be conservative (c.f. Supplementary Table 10 in the Supplementary Note and reproduced below for reference), and find that a Bonferroni correction is only very slightly conservative i.e.:

“The one-degree-of-freedom strategies investigated in this manuscript involve double the number of statistical tests across the genome as the two-degree-of-freedom strategy. We address this issue by dividing the significance threshold for the one-degree-of-freedom strategies by two (i.e. adjusting the threshold for genome-wide significance from $\alpha = 6.6 \times 10^{-9}$ to $\alpha = 3.3 \times 10^{-9}$). To investigate the appropriateness of this procedure, we simulated the distribution of the maximum chi-square statistic (i.e. from potentially correlated fetal and maternal tests of association) under the null hypothesis of no association (1 million draws) whilst varying the sample overlap (we assume 1000 individuals in the maternal test, and 1000 individuals in the fetal test) and the correlation between maternal and fetal phenotype. We calculate the quantile corresponding to the upper 5%, 1% and 0.1% tail of this maximum chi-square distribution. We then determine what upper tail probability of the one-degree-of-freedom central chi-square distribution this value corresponds to (Supplementary Table 1). Our results suggest, that even in the case of 100% overlap and $\rho = 0.5$, a Bonferroni correction is not too conservative.”

Supplementary Table 10 Upper tail probabilities of the one-degree-of-freedom central chi-square distribution associated with the corresponding quantile from the maximum chi-square distribution.

	$\alpha = 0.05$	$\alpha = 0.01$	$\alpha = 0.001$
0% overlap, $\rho = 0$	0.0253	0.00500	0.000496
50% overlap, $\rho = 0$	0.0251	0.00503	0.000485
100% overlap, $\rho = 0$	0.0252	0.00505	0.000484
0% overlap, $\rho = 0.5$	0.0253	0.00498	0.000487
50% overlap, $\rho = 0.5$	0.0256	0.00503	0.000501
100% overlap, $\rho = 0.5$	0.0269	0.00525	0.000536

To accommodate the reviewer’s suggestion, we also examined a strategy that integrates p-values across the different tests without correction due to multiple testing. Specifically, we examined the performance of the “Cauchy combination method” (Liu et al. 2019 *Am J Hum Genet*), which is implemented in several recent genetic software package including MBAT-Combo (Li et al. 2023 *Am J Hum Genet*), on power to detect maternal and fetal genetic effects.

Briefly, we simulated bivariate normally distributed quantitative traits of unit variance for own and offspring phenotype that were influenced by a single additive locus (N = 1000 simulations). We varied maternal and fetal effect sizes ($\beta_{\text{f}} = -0.1, -0.05, 0, 0.05, 0.1$ and $\beta_{\text{m}} = -0.1, -0.05, 0, 0.05, 0.1$), residual correlation between maternal and fetal phenotype (three conditions: (i) $\rho = 0$; (ii) $\rho = 0.25$; (iii) $\rho = 0.5$), sample size of each GWAS (19

conditions: $N = 100$ to 900 , increment by 100 , and then $N = 1,000$ to $10,000$, increment by 1000), relative sample size (two conditions: (i) equal sized maternal and fetal GWAS; (ii) maternal GWAS half the size of fetal GWAS) and extent of sample overlap (three conditions: (i) zero overlap; (ii) 50% of individuals in the maternal GWAS in the fetal GWAS; (iii) all of the mothers in the maternal GWAS in the fetal GWAS). We then compared the power estimated using the best p-value from maternal or fetal one-degree of freedom test ($\alpha = 3.3 \times 10^{-9}$) against power estimated using the Cauchy combination test of the two statistics ($\alpha = 6.6 \times 10^{-9}$). The results show the two strategies provide very similar power- in other words, there is little to be gained by employing this strategy, and so we do not examine it further in the manuscript.

Unconditional 1df test

3. I think an evaluation of type 1 error is necessary. The supplementary figure 1 shows that the overall distribution of test statistics is not significantly different from the theoretical one. However, in large-scale genetic studies, what matters most is to have accurate type I errors at the tail of distribution (e.g., $<5 \cdot 10^{-8}$).

We investigate type I error rate in the tail end of the distribution using permuted data in the UK Biobank (Supplementary Table 1). We also now include estimates of type I error (i.e. when $\beta_m = 0$ and $\beta_f = 0$) in Supplementary Table 2. Type I error is well controlled in all situations.

4. At the end of abstract, the author concluded that “a simple meta-analytic strategy is likely to perform best, particularly in situations where maternal and fetal GWAS only partially overlap.” I am not sure this is reflecting the simulation and application results in Fig 1 and Fig 2. Fig 1 is difficult to read (maybe better color and legend can help). However, it is difficult for me to conclude from this figure that meta-analytic has meaningful improvement over 2-df test. It appears that the meta-analytic may have slightly improved power, which does not seem significant, while may suffer substantial power loss when the effects are bi-directional. Figure 2 also seems to suggest 2-df has improved power with smaller p-values.

The reviewer is looking at the wrong figure on which our conclusions are based. The relevant comparison between the one-degree-of-freedom meta-analysis strategy and the two-degree-of-freedom strategy is contained in Figure 6 not Figure 2 (i.e. which compares the two-degree-of-freedom strategy against the traditional strategy of performing separate maternal and fetal GWAS). Figure 6d shows that the meta-analytic strategy captures 21 loci which are not significant in the two-degree-of-freedom test, whereas the two degree of freedom test only captures 8 loci not significant using the meta-analytic strategy.

The relative performance of the two strategies is discussed extensively in the section on “Empirical analyses of birth weight using one-degree-of-freedom meta-analyses” in the Results section:

“Empirical analyses of birth weight using one-degree-of-freedom meta-analyses

The one-degree-of-freedom meta-analytic strategy identified 369 independent SNPs at 226 loci that were genome-wide significant ($p\text{-value} < 3.3 \times 10^{-9}$) in either the meta-analysis of fetal effects, or the meta-analysis of maternal effects – an even greater number of loci than were identified using the two-degree-of-freedom strategy (Supplementary Table 6). These included 21 independent genome-wide significant SNPs at 21 loci that were not identified in the two-degree-of-freedom T_{2df} tests (Figure 6, Supplementary Table 6), although all only narrowly missed out on significance in the two-degree-of-freedom T_{2df} test, having a p-value smaller than 5×10^{-8} . Interestingly, eight independent SNPs (at eight loci) were significant in the T_{2df} test but not in the meta-analysis of fetal effects, or the meta-analysis of maternal effects (Figure 6, Supplementary Table 3, Supplementary Table 6). Four of these SNPs exhibited significant ($p\text{-value} < 0.05$) discordant maternal and fetal effects in the conditional analyses– most notably rs7034200 at *GLIS3*, a known type 2 diabetes locus, that was highly significant in the two-degree-of-freedom test ($p\text{-value}_{2df} = 1.15 \times 10^{-12}$), but only exhibited relatively modest non-significant p-values in the maternal and fetal meta-analyses (minimum $p_{meta} = 7.39 \times 10^{-5}$), and likewise rs560887 at *G6PC2*, a known locus for fasting glucose which also exhibited radically different p-values across the two tests ($p\text{-value}_{2df} = 4.37 \times 10^{-20}$, minimum $p_{meta} = 6.25 \times 10^{-7}$).”

and also in the Discussion:

“So which type of analysis is optimal in the case of birth weight and (potentially) other perinatal phenotypes, particularly in situations where there is only partial overlap between maternal and fetal GWAS? The meta-analytic strategy yielded considerably greater numbers of loci meeting genome-wide significance than the two-degree-of-freedom strategy (i.e. 226 vs 213 loci), suggesting that even for a trait like birth weight, a simple meta-analytic strategy may be optimal when maternal and fetal GWAs are only partially overlapping. However, whilst all loci that met genome-wide significance using the simple meta-analytic strategy were also significant or almost significant using DINGO (i.e. all loci would be flagged by researchers as significant or suggestive), many genome-wide significant loci that displayed directionally discordant effects in the two-degree-of-freedom DINGO test exhibited p-values that were far less extreme using the simple meta-analytic strategy, including some loci that may have been missed *GLIS3* locus discussed below). The implication is that a two-degree-of-freedom test may be the preferable strategy in the case of phenotypes that are known to exhibit substantial numbers of loci with directionally discordant effects (e.g. perinatal growth-related traits like birth weight), whereas a simple meta-analytic strategy may be the superior strategy in the case of other traits (bearing in mind that such knowledge about likely underlying genetic architecture may not always be available from previous GWAS *a priori*). We have implemented both the meta-analytic tests and the two-degree-of-freedom test as part of the DINGO package in the publicly available web-based software CTG-VL¹⁷.”

To improve clarity, we have:

-Relabelled Figure 2 to emphasize that this comparison is between the two degree of freedom strategy and the traditional strategy of conducting separate maternal and fetal GWAS.

-Coloured the datapoints in Figures 6a-6c where the meta-analytic strategy outperforms the two-degree-of-freedom strategy. We have also added additional text to the Figure legend that helps explain the relative merits of the two strategies.

5. Would it be reasonable to conduct 1-df meta-analytics for the coefficients estimated from separate GWAS similar to the proposed strategy? Again, if the goal is truly gene discovery, would it have similar performance though the effect estimates are biased?

No because such a strategy does not take sample overlap into account (i.e. the type I error rate would be inflated at loci that are not related to the phenotype).

6. I am confused with the descriptions of sample overlaps, information overlap, independence, etc. It seems that the authors use 100% sample overlap for studies with mother-offspring pairs. I think I understand what you are trying to say, but it does not seem correct to describe it as 100% sample overlap. There are overlapping information also independent information.

We agree that the issue of sample overlap has the potential to be very confusing! To be clear, we do not describe mother-offspring pairs as having 100% overlap, but rather 50% overlap. The effective degree of sample overlap is described on page 4 of the manuscript:

“ N_s is the effective number of overlapping individuals across both GWAS (e.g. in situations where all mothers report their own and their offspring’s birth weight the effective sample overlap would be 100%, whereas for genotyped mother-offspring pairs where only the offspring’s phenotype is measured, the effective sample overlap would be 50%, since offspring and maternal genotypes are correlated 0.5- these situations are illustrated in Supplementary Figure 1),”

Given the confusion, and in order to make this clearer, we have created a new Supplementary Figure that describes some of the possible patterns of sample overlap that could arise.

7. The method, as currently proposed, is for quantitative phenotypes, which needs to be reflected in the title and abstract.

We now include the word “quantitative” in the title of the manuscript:

“Direct and Indirect effects analysis of Genetic IQci (DINGO): A software package to increase the power of locus discovery in GWAS meta-analyses of perinatal quantitative phenotypes and traits influenced by indirect genetic effects”.

and also in the abstract:

“In this manuscript we investigate the performance of three strategies to detect loci in maternal and fetal GWASs of the same quantitative trait:...”

8. The authors discussed the potential extension to 3-df test with paternal effect. It is hard to believe that paternal effect will function without transmitting the risk allele (fetal effect). Please provide references if such examples exist.

Educational attainment shows robust evidence for paternal genetic effects (e.g. Wang et al. 2021 *Am J Hum Genet*).

We have modified the following text of the manuscript:

“With a couple of notable exceptions (e.g. the Norwegian MOBA cohort), the majority of the world’s birth cohorts contain only limited genotype information on the children’s fathers. Nevertheless, in some situations it may be useful to include information from a GWAS of offspring phenotype regressed on father’s genotype. This could be useful when paternal effects are suspected to contribute to offspring trait variation (e.g. in the case of educational attainment (Wang et al. 2021))...”

9. Minor comments. There are other confusing descriptions and errors impact the understanding.

- “own genotype” and “own phenotype” are used to refer either maternal or fetal data at different places is confusing. I suggest being consistent with maternal GWAS and fetal GWAS.

We now consistently use “own phenotype on own genotype” to describe the fetal GWAS, and “offspring phenotype on maternal genotype” to describe the maternal GWAS.

- Line 48, should read “due to multiple testing”.

Changed.

- “two-degree” instead of “two degree”

We have changed all instances of two degree of freedom to two-degree-of-freedom, and all instances of one degree of freedom to one-degree-of-freedom throughout the manuscript and supplementary materials.

- “information shared across the individual GWAS” and “information contained across both GWAS” at different places. Also, the use of “GWAS” as both single and plural forms.

We have deleted the second of these sentences. We now use GWAS (singular) and GWASs (plural).

- Line 50, the sentence is problematic.

“Data simulations and asymptotic power calculations we have performed previously using a computationally intensive structural equation model (i.e. that is not suitable for GWAS) have hinted at the gains in power that might be achieved by modelling and testing indirect and direct effects simultaneously in one analysis.”

We have simplified the sentence to:

“Data simulations and asymptotic power calculations we have performed previously have hinted at the gains in power that might be achieved by modelling indirect and direct effects simultaneously.”

- Explain abbreviations, such as MTAG.

MTAG stands for Multi-trait analysis of GWAS. We now define this abbreviation the first time it is used in the manuscript.

REVIEWER COMMENTS

Reviewer #1 (Remarks to the Author):

1. Supplementary Figures 4 and 6: I can only see the blue lines but not the other two? Maybe the authors can recreate the figures to make the results of the other two methods visible?

We recreated the figures to make the results of the other two methods visible.

2. It's hard to see which ones are type one error rates in Supplementary Table 2. I'm also confused about what is the level of type I error control here. It seems like when $\beta_m = 0$ and $\beta_f = 0$, all p-values are 0?

P-values are now displayed in the "Scientific format" instead of "General format" in the excel spreadsheet to avoid small numbers being shown as 0. Type 1 errors are also plotted in Supplementary Figures 3-7 in the right column where $\beta_m = 0$ and $\beta_f = 0$.

Reviewer #1 (Remarks on code availability):

The authors have provided the source code of the DINGO software but not the code that was used to generate the results presented in the manuscript.

R code used to generate the results is now included at the end of the Supplementary Note.

Reviewer #2 (Remarks to the Author):

The authors have responded to my comments.

Reviewer #2 (Remarks on code availability):

R codes are provided. A readme file would be helpful.

A readme file is now provided at:
https://gist.github.com/daniellhdwang/f6485dd4ebcf0d72f771804cc0f6f678?permalink_comment_id=5108161#gistcomment-5108161

README

GitHub Gist: instantly share code, notes, and snippets.
gist.github.com

REVIEWER COMMENTS

Reviewer #1 (Remarks to the Author):

The code looks fine. However, I could not check the reproducibility of the results because no data was provided.

Links to summary results data for the deCODE maternal and fetal GWAS of birthweight, as stated clearly in the data availability section of our previous revision, are located at <https://www.decode.com/summarydata/>.

The reviewer will find the relevant links on this website under the 2021 section and the Juliusdottir et al paper:

https://download.decode.is/form/2021/Birthweight_offspring_mothers2021.gz
<https://download.decode.is/form/2021/Birthweight2021.gz>

The user will be directed to a form that sets out the conditions of use and requires users to electronically sign in order to download the data.

We have made this even more explicit in an update to our data availability section:

Data Availability

GWAS summary results statistics of birth weight published by deCODE are available at <https://www.decode.com/summarydata> and can be downloaded here <https://download.decode.is/form/2021/Birthweight2021.gz> and here https://download.decode.is/form/2021/Birthweight_offspring_mothers2021.gz. GWAS summary results statistics from the DINGO analysis of birth weight are available at the Evans Group website (<https://evansgroup.di.uq.edu.au/>) upon publication. The source code of the DINGO software is available at <https://gist.github.com/daniellhdwang/1f8512224e2979bf436facd6f966b05a>.

These datafiles are very large and we do not have authority to distribute them. In addition, there are some formatting issues with the files that requires some preprocessing- which we have also outlined previously in the “Empirical Application” section of the Methods of our paper. If the reviewer wants to check the reproducibility of our code, we have included a zip file with four additional files (not for publication):

Here we provide four files for Reviewer 1 to check the reproducibility:

1. GWAS of own birth weight for the 369 SNPs identified in the 1df meta-analysis (See Supplementary Table 6).
2. GWAS of offspring birth weight for the 369 SNPs identified in the 1df meta-analysis (See Supplementary Table 6).
3. R_scripts to run the DINGO analysis using scripts provided in the Supplementary Note.
4. Outputs generated by the R_scripts, which contains 2df and 1df-meta-analysis results as shown in the Supplementary Table 6.